# Explanatory Instructions:
# Towards Unified Vision Tasks Understanding and Zero-shot Generalization

**Yang Shen** [1]  **Xiu-Shen Wei** [2 †]  **Yifan Sun** [3 †]  **Yuxin Song** [3]  **Tao Yuan** [3]  **Jian Jin** [1]  **Heyang Xu** [4]  **Yazhou Yao** [1 5]
**Errui Ding** [3]

## Abstract

Computer Vision (CV) has yet to fully achieve the zero-shot task generalization observed in Natural Language Processing (NLP), despite following many of the milestones established in NLP, such as large transformer models, extensive pre-training, and the auto-regression paradigm, among others. In this paper, we rethink the reality that CV adopts discrete and terminological task definitions (*e.g.*, "image segmentation"), and conjecture it is a key barrier that hampers zero-shot task generalization. Our hypothesis is that without truly understanding previously-seen tasks–due to these terminological definitions–deep models struggle to generalize to novel tasks. To verify this, we introduce Explanatory Instructions, which provide an intuitive way to define CV task objectives through detailed linguistic transformations from input images to outputs. We create a large-scale dataset comprising 12 million "image input $\rightarrow$ explanatory instruction $\rightarrow$ output" triplets, and train an auto-regressive-based vision-language model (AR-based VLM) that takes both images and explanatory instructions as input. By

learning to follow these instructions, the AR-based VLM achieves instruction-level zero-shot capabilities for previously-seen tasks and demonstrates strong zero-shot generalization for unseen CV tasks. Code and dataset have been openly available on our GitHub repository.

## 1. Introduction

Natural Language Processing (NLP) has achieved significant advancements in task-level zero-shot generalization, driven by key milestones such as transformer models, large-scale pretraining, and the auto-regression paradigm (Vaswani et al., 2017; Brown et al., 2020; Touvron et al., 2023a). This impressive adaptability stems from the language models' capacity to interpret flexible, open-ended but semantically rich instructions, enabling them to generalize across language tasks without additional task-specific fine-tuning.

In contrast, Computer Vision (CV) has not yet unlocked the task-level zero-shot generalization capabilities. Early CV models largely depend on discrete vision task boundaries (cf. Fig. 1 (a)), which restricts their flexibility and generalization capabilities. While recent efforts, *e.g.*, Lumina-mGPT (Liu et al., 2024), OmniGen (Xiao et al., 2024), and PixWizard (Lin et al., 2024), have attempted to incorporate a wide range of vision tasks via fixed or open-style terminological instructions (cf. Fig. 1 (b)), their zero-shot capabilities remain confined within seen vision tasks or rely on few-shot prompting for guidance. Our intuition strongly implies that this limitation may stem from the constrained scope of instructions and existing vision-language models (VLMs) do not truly understand the previously-seen vision tasks. To be specific, these terminological instructions are consisted of discrete and symbolic task definitions that do not convey the objective behind each task, which can be replaced with any other symbols (*e.g.*, changing "Semantic segmentation" within these instructions into "Task X" or "Task djjjvau") without affecting the vision task results.

To enable deep models to understand vision tasks genuinely, we introduce Explanatory Instructions that characterize vision tasks by providing linguistic descriptions of the task

---

[1]School of Computer Science and Engineering, Nanjing University of Science and Technology, China. [2]School of Computer Science and Engineering, and Key Laboratory of New Generation Artificial Intelligence Technology and Its Interdisciplinary Applications, Southeast University, China. [3]Baidu VIS. [4]School of Instrument Science and Engineering, and State Key Laboratory of Digital Medical Engineering, Southeast University, China. [5]State Key Laboratory of Intelligent Manufacturing of Advanced Construction Machinery, China. This work was supported by National Key R&D Program of China (2021YFA1001100), National Natural Science Foundation of China under Grant (62272231, 62472222), CIE-Tencent Robotics X Rhino-Bird Focused Research Program, the Fundamental Research Funds for the Central Universities (4009002401), Natural Science Foundation of Jiangsu Province (BK20240080), and the Big Data Computing Center of Southeast University. Correspondence to: Xiu-Shen Wei <weixs.gm@gmail.com>, Yifan Sun <sunyf15@tsinghua.org.cn>.

*Proceedings of the 42nd International Conference on Machine Learning*, Vancouver, Canada. PMLR 267, 2025. Copyright 2025 by the author(s).

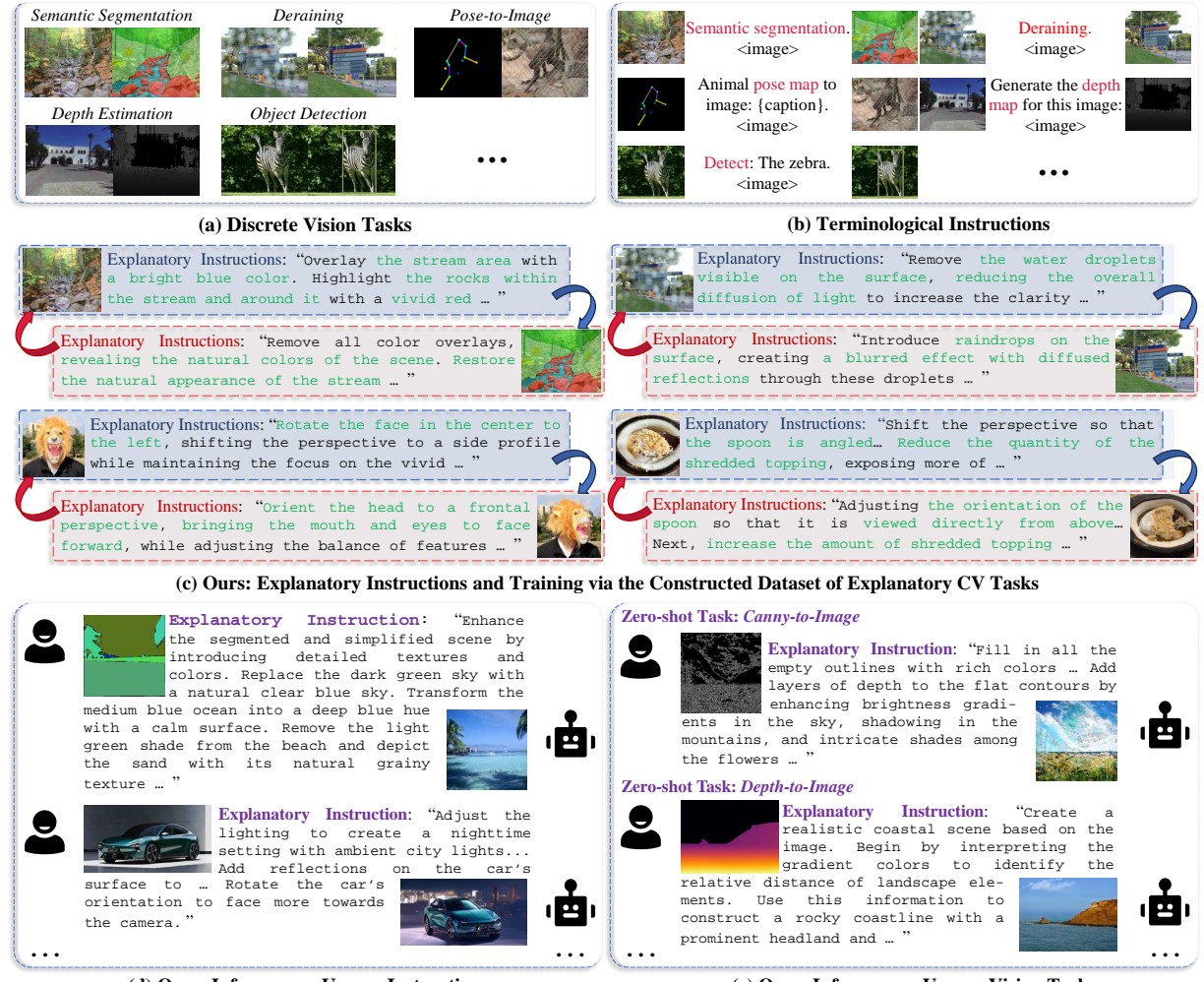

*Figure 1.* (a) Early CV models are designed to handle discrete vision tasks. (b) Recent VLMs use terminological instructions (*i.e.*, terminological task definitions), *e.g.*, "semantic segmentation" and "pose map". (c) We propose Explanatory Instructions to explain CV tasks' objective and construct the dataset of Explanatory CV Tasks. We train the model via this dataset. (d) The trained model showcases instruction-level zero-shot capabilities. (e) By omitting certain human-defined vision tasks in the training dataset (cf. Sec. 4.2), we demonstrate promising vision task-level zero-shot capabilities.

objective, *i.e.*, the detailed transformations from input images to outputs, cf. Fig. 1 (c). To provide context, we first revisit the concept of vision tasks. Generally, a vision task refers to a specific objective or operation within the visual domain that humans or machines need to accomplish (Marr, 2010). However, current vision tasks are constrained by rigid, human-defined boundaries, *e.g.*, controllable generation tasks such as *Depth-to-Image*, *Canny-to-Image*, *Pose-to-Image*, dense image prediction tasks such as *Segmentation*, *Pose Estimation*, *Surface Normal Estimation*. In practice, researchers present vision tasks to models via discrete and terminological definitions, limiting both the articulation of task objectives and the expressive flexibility of task descriptions. For instance, when describing a segmentation task, non-experts often use natural, descriptive phrases that, more importantly, carry clear and specific meanings, *e.g.*, "overlay the stream area with a bright blue color" or

"paint the stream in blue and mark the rocks within it in red", rather than narrowly defined terminological instructions such as "semantic segmentation" or close variants.

Through this form of instruction, we clarify the true objectives of various vision tasks, moving them beyond the constraints of terminological task categories. We further help enhance the diversity of task expressions within the vision domain, while also strengthening the alignment between text and vision modalities at the task level. To operationalize this idea, we construct the Dataset of Explanatory Computer Vison Tasks (DECVT) that contains 12 million "image input → explanatory instruction → output" triplets. Notably, for the same image inputs, DECVT provides different explanatory instructions and corresponding output images for varying task objectives. Furthermore, DECVT adopts

explanatory instructions to represent transformations in both directions, *e.g.*, from original images to segmentation outputs and from segmentation outputs back to original images (cf., the segmentation example in Fig.1 (c)), capturing a richer diversity of task expressions. By omitting certain terminological-based vision tasks, *e.g.*, *Canny-to-Image* and *Image-to-Canny*, *Depth-to-Image* and *Image-to-Depth*, we conduct instruction-driven supervised fine-tuning on an auto-regressive-based vision-language model (AR-based VLM), utilizing approximately 1.5 million bidirectional pair of "image ↔ explanatory instructions ↔ image" triplets (*i.e.*, input image, output image, explanatory instructions from input to output and from output to input). Results show that the fine-tuned model exhibits both instruction-level zero-shot capabilities (cf. Fig. 1 (d)) and promising task-level zero-shot capabilities (cf. Fig. 1 (e)), demonstrating its potential to generalize effectively across diverse vision tasks.

The main contributions of this work are as follows:

• We propose Explanatory Instructions that intuitively explain task objectives by characterizing diverse vision tasks through linguistic descriptions of the transformations between paired images. We further construct a Dataset of Explanatory CV Tasks, which, to the best of our knowledge, is the first dataset to integrate multiple vision tasks while providing instructions that authentically describe the task objectives.

• We conduct experiments to demonstrate that, after learning to follow the proposed Explanatory Instructions, the AR-based VLM model exhibits zero-shot generalization not only at the instruction level but also at the vision task level. Besides, fine-tuning on the constructed dataset further enhances the model's versatility, enabling it to handle any combination of tasks, rather than being restricted to a single task that relies on specific terminological instructions. This advancement represents a significant step toward a more flexible and unified understanding of vision tasks.

## 2. Dataset of Explanatory CV Tasks

We construct a Dataset of Explanatory CV Tasks (DECVT) based on our proposed Explanatory Instructions fashion, where vision tasks are represented exclusively through textual descriptions of transformations between images, effectively clarifying each task's objective without relying on predefined terminological categories. During the construction of DECVT, we consider the discrete nature of the established concept of computer vision tasks and divide the dataset into two components, including "Terminological-based Vision Tasks" and "Explanatory-based Vision Tasks". The first component primarily includes tasks whose objectives have been abstracted into standardized terminologies, while the second component encompasses tasks that require explanatory descriptions to differentiate them. These

**Caption of image A:**
A close-up of a traditional moon-cake with an intricate, embossed design on its golden-brown crust, placed on a decorative white doily.

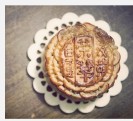

**Caption of image B:**
A simplified version of the moon-cake where only the main shapes and contours of the design are visible, showing a stark contrast between the outlines and the background, with all detailed textures removed.

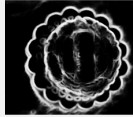

**Explanatory instruction from A to B:**
"Reduce the detailed visual elements, focusing only on the essential shapes and contours of the design. Strip away color and texture, retaining only the prominent outlines to emphasize the primary form and structure against a plain background."

**Explanatory instruction from B to A:**
"Add layers of color, texture, and depth to the outlined shapes, recreating the intricate details and embossed patterns. Introduce a warm, golden-brown hue to bring out the richness of the surface, restoring the complete and detailed appearance of the object with its decorative base."

*Figure 2.* Examples of terminological-based vision tasks, *e.g.*, holistically nested edge detection.

two components contain approximately 6 million bidirectional pairs of "image ↔ explanatory instructions ↔ image" triplets (*i.e.*, around 12 million individual "input image → explanatory instruction → output" triplets). Below, we provide an introduction along with example entries for each component. Please refer to Appendix A for specifics on the dataset construction.

### 2.1. Terminological-based Vision Tasks

This component primarily encompasses low-level vision tasks, controllable generation tasks, dense image prediction tasks, image grounding tasks and their inverse counterparts. While these tasks have been refined and succinctly abstracted through human summarization, their inherent discreteness limits the model's ability to comprehend these vision tasks fully. Therefore, we aim to reduce the rigidity by providing detailed explanatory instructions that explicitly articulate the underlying objectives of each task. To further explore task-level zero-shot generalization capabilities, this component includes only a subset of terminological-based vision tasks. Specifically, we incorporate low-level data from open-source datasets including AllWeather (Valanarasu et al., 2022), NYU-Rain (Li et al.,

2019b), D-HAZE (Ancuti et al., 2016), NH-HAZE (Ancuti et al., 2020), UIEB (Li et al., 2019a), Adobe-5k (Bychkovsky et al., 2011), covering restoration tasks of *Image Restoration*, *Deraining*, *Dehazing* and *Desnowing*. We also collect *Object Detection* data from the LVIS (Gupta et al., 2019) dataset, *Style Transfer* data from CSGO[1](Xing et al., 2024) and prepare data for dense image prediction tasks including *Depth Estimation*, *Surface Normal Estimation*, *Pose Estimation*, and *Semantic Segmentation* from ADE20K (Zhou et al., 2017b), Depth in the Wild (Chen et al., 2016) and randomly selected data from the next two components. Additionally, for controllable generation tasks, we incorporate *Holistically-Nested Edge (HED) Boundary to Image* and inverse dense image prediction tasks (*e.g.*, *Pose-to-Image*, *Depth-to-Image*, *Segmentation-to-Image*). Examples for this component are shown in Fig. 1 (c) and Fig. 2. Details and additional examples are provided in Appendix A.2 and Appendix A.3.2.

## 2.2. Explanatory-based Vision Tasks

Compared to language tasks, vision tasks inherently require more intricate forms of expression. While leveraging terminological-based vision tasks offers a solid foundation, their predefined boundaries can limit the diversity and complexity of task objectives. To address this, we introduce explanatory-based vision tasks, which utilize descriptive instructions to articulate the transformations between images, enabling a broader and more diverse range of vision task objectives.

We first collect data related to instruction-based image editing, where diverse editing instructions enable users to perform straightforward edits using natural language commands. However, each editing instruction typically involves only a single operation of modification, such as adding, removing, or modifying the background or specific objects within an image, while preserving the overall structure and contextual integrity of the unaltered regions. Specifically, we directly incorporate publicly available image editing datasets including MagicBrush (only involving the real world and multi-turn subsets) (Zhang et al., 2023a), SEED (Ge et al., 2024), HQ-Edit (Hui et al., 2024), HIVE (Zhang et al., 2024), InstructPix2Pix (Brooks et al., 2023) and PromptFix (Yu et al., 2024).

Afterward, we seek to transcend established vision task paradigms by incorporating more visual-related image pairs, thereby expanding the scope of task objectives. The transformations between these image pairs involve sophisticated variations which cannot be adequately described with straightforward instructions that typically focus on individual changes, *e.g.*, combinations of camera angles with intri-

---

[1]As CSGO has not been open-sourced, we only collect data from the paper.

**Caption of image A:**
A glass of iced latte with a light creamy color sits on a wooden surface. The drink has visible ice cubes floating at the top, creating a refreshing appearance. The background is softly blurred, focusing the attention on the drink and enhancing its warm, cozy aesthetic.

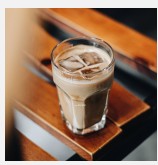

**Caption of image B:**
A glass of iced coffee with distinct layers, where dark espresso swirls into the lighter creamy base. The drink is set on a hexagonal coaster placed on a textured glass surface, with a blurred, minimalistic background. The presentation emphasizes the richness of the espresso and gives a modern, refined feel to the scene.

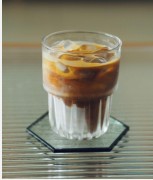

**Explanatory instruction from A to B:**
"Change the wooden surface beneath a chilled drink presentation to a glossy, textured platform that refracts light. Alter the arrangement and appearance of the beverage to have a swirl of ingredients visible from the side, creating a smoother gradient effect. Adjust the lighting to soften shadows and introduce a more ambient illumination across the scene, highlighting the layers and making the glass's ridges appear more prominent."

**Explanatory instruction from B to A:**
"Replace the sleek, reflective surface beneath the clear beverage vessel with a warm, rustic wooden texture that enhances the natural, earthy tone of the scene. Modify the beverage's visual layers to achieve a more blended appearance with a focus on a single color. Adjust lighting conditions to create sharper contrasts and distinct edges."

*Figure 3.* Examples of explanatory-based vision tasks.

cate lighting compositions, nuanced adjustments to scene composition and visual ambiance, or macro-level visual content changes but with similar features. These complex dynamics extend beyond the capabilities of conventional terminological-based vision tasks and require more intricate language instructions. We collect images to expand this component from search engines, with examples shown in Fig. 1 (c) and Fig. 3. Details on constructing these explanatory instructions and additional examples are provided in Appendix A.1 and Appendix A.3.1.

## 3. Methodology

The primary objective of this work is to demonstrate that VLMs can achieve both instruction-level and vision task-

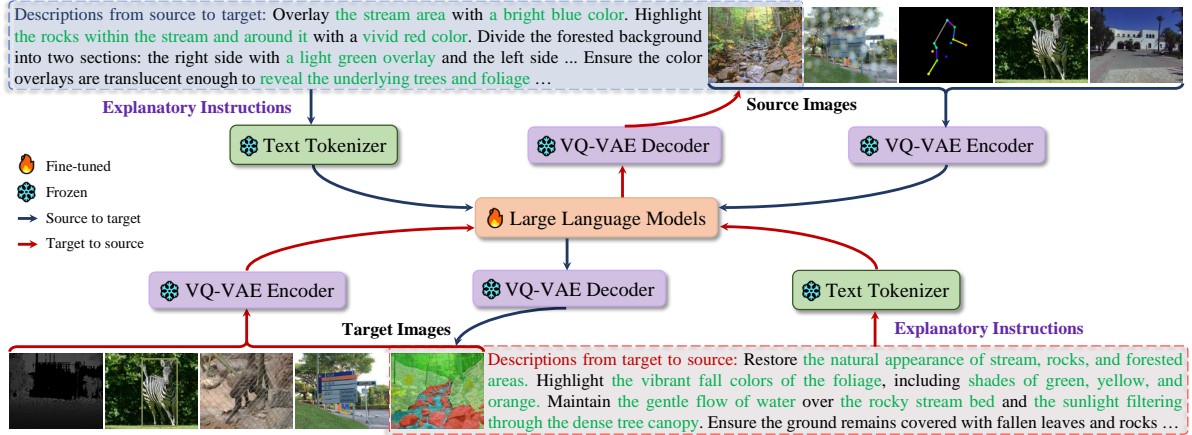

*Figure 4.* Framework of our vanilla token-based VLM method.

level zero-shot capabilities using the proposed Explanatory Instructions, which characterize vision tasks through linguistic descriptions of their objectives, rather than to provide a state-of-the-art solution. Consequently, we adopt a straightforward token-based VLM that integrates an image tokenizer/de-tokenizer with a general mixed-modal AR model. The overall framework is illustrated in Fig. 4.

**Architecture** To create a unified token sequence, both text and images should be tokenized into discrete spaces. Following Chameleon (Team, 2024) and Lumina-mGPT (Liu et al., 2024), we employ a byte pair encoding tokenizer (Sennrich, 2015) for the text modality and use a quantization-based tokenization method to convert continuous image patches into discrete tokens for the image modality (van den Oord et al., 2017; Razavi et al., 2019; Esser et al., 2021; Ramesh et al., 2021). After projecting text and image inputs into a unified sequence by concatenating the text tokens and the flattened 1D image tokens, we employ a decoder-only autoregressive transformer utilizing the standard dense transformer architecture to simplify the generative modeling. Following LLaMa-2 (Touvron et al., 2023b), we adopt adaptations including RMSNorm (Zhang & Sennrich, 2019) for normalization, SwiGLU (Shazeer, 2020) for activation and rotary positional embeddings (RoPE) (Su et al., 2024) for positional encoding. To enhance training stability, Pre-LayerNorm (Xiong et al., 2020), Post-LayerNorm (Ding et al., 2021) and Query-Key Normalization (QK-Norm) (Henry et al., 2020) are incorporated into each transformer block. Unambiguous Image Representation (Uni-Rep) (Liu et al., 2024) are added to enable image generation at flexible resolution and aspect ratio.

**Initialization** The core contribution of this work lies in redefining vision tasks through Explanatory Instructions and investigating their potential for generalization, shifting the focus away from pretraining a decoder-only autoregressive transformer from scratch. To achieve this, we leverage a multimodal generative model pretrained on large-scale image-text datasets, which provides flexible image gener-

ation capabilities. This approach eliminates the need for random initialization or language-only model initialization, conserving substantial computational resources otherwise required for establishing foundational image generation capabilities. Consequently, we can concentrate on exploring zero-shot generalization at both the instruction level and the vision task level within a vanilla token-based VLM.

**Supervised Fine-tuning** During training, the decoder-only auto-regressive transformer models the conditional probability $p(x_t|x_1, x_2, ..., x_{t-1})$ of multimodal sequences using the standard next-token prediction objective. Following previous works (Chowdhery et al., 2023; Wortsman et al., 2023), we apply $z$-loss to prevent the model's output logits from becoming overly confident, thereby enhancing generalization. To streamline the supervised fine-tuning process, we organize all the data into single-dialog formats (*e.g.*, for multi-turn editing data, if image $\mathcal{I}_A$ is edited to image $\mathcal{I}_B$ by instruction $\alpha$ and then to image $\mathcal{I}_C$ by instruction $\beta$, we treat this as a single sequence where image $\mathcal{I}_A$ is edited into image $\mathcal{I}_C$ by instruction $\alpha + \beta$), applying the loss function exclusively to output tokens. For all the experiments, we employ the AdamW (Loshchilov, 2017) optimizer with a weight decay of 0.01 and betas set to (0.9, 0.95). The learning rate is configured at $4 \times 10^{-5}$, and the $z$-loss is applied with a weight of $10^{-5}$. To increase training throughput, all the data are pre-tokenized before training and clustered based on the number of tokens.

**Inference** During inference, any description associated with the input image can be provided as an explanatory instruction, guiding the fine-tuned token-based VLM to generate the corresponding output image based on both the image and the instruction. Furthermore, the sampling strategy for auto-regressive models involves various hyper-parameters that significantly affect the results. For example, while the top-$k$ value is typically set to 5 for text generation in LLMs, we recommend increasing the top-$k$ to 2048 for the image generation stage.[2]

---

[2] More discussions are provided in Appendix D

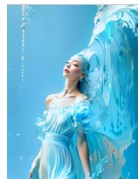

**Explanatory Instruction**: "`Transform the water elements into fire elements, using a burning flame as the main theme. The hair becomes flame-like, in orange and red tones, rising upwards with subtle sparks. The dress is redesigned with flowing flame textures, resembling burning silk, creating dynamic contrasts between light and shadow. The background shifts to resemble molten lava, with a lively fire flicker.`"

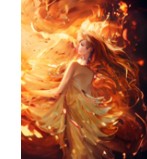

**Input Image**

**Output Image**

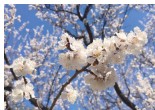

**Explanatory Instruction**: "`Dim the scene's brightness, reduce the intensity of sunlight, and adjust the environment to a rainy, overcast setting. Add a sense of light drizzle and mist to the scene, creating a soft yet slightly melancholic atmosphere. Gradually increase the vibrancy of the pink tones around the petals, but give them a faintly faded and blurred appearance, and introduce a sense of decay.`"

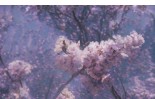

**Input Image**

**Output Image**

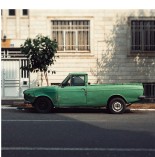

**Explanatory Instruction**: "`Change the color of the car to red while try to keep all other objects in the scene unchanged. Add a layer of snow to the surfaces of the road, the car, and the trees to create the atmosphere of a winter morning. Pay close attention to details such as shadows and lighting adjustments.`"

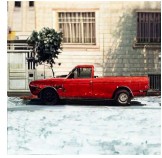

**Input Image**

**Output Image**

*Figure 5.* Examples of instruction-level zero-shot capabilities.

# 4. Experiments

In this section, we evaluate the model's zero-shot capabilities on unseen instructions and unseen vision tasks through different experimental settings.

## 4.1. Zero-shot Capabilities on Unseen Instructions

**Settings**   We fine-tune a 7B AR-based VLM by utilizing all the DECVT training data. The pre-trained model used to initialize is "Lumina-mGPT-7B-768-Omni". Training is conducted on 64 A100 GPUs, with batch size for each GPU set as 8 over 2 epochs (around 47k iterations), totaling 5,400 GPU hours. Image resolution equals $448 \times 448$.

**Instruction-level Zero-shot Capabilities**  Certain terminological-based tasks in NLP, *e.g.*, *Machine Translation*, *Emotion Detection*, *Named Entity Recognition*, and *Relation Extraction*, have clearly defined task boundaries (*i.e.*, their objectives are specific). However, many other language tasks, such as *Dialogue* and *Question Answering*, lack such clear task objective boundaries. Similarly, we argue that the set of terminological-based vision tasks represents only a small subset of possible vision-related tasks. Many vision-related tasks are likely analogous to *Dialogue* in NLP, lacking clear task definitions or boundaries. As a result, it is difficult to determine whether certain vision tasks are explicitly included during dataset construction (just as dataset for LLMs often include *Dialogue* or *Question Answering* data). However, it is easy to calculate textual similarity to verify whether the instructions associated with a given image are part of the training set. Therefore, we refer to the model's ability to generalize on such instructions and images as instruction-level zero-shot capabilities. Examples in Fig. 5 provide evidence that the AR-based VLM fine-tuned with the proposed explanatory instructions exhibits promising

potential to become a versatile vision generalist. Note that, in addition to image editing examples, our proposal also performs well on traditional discrete vision tasks, such as *Semantic Segmentation*, *Surface Normal Estimation*, and more, as detailed in Appendix C.

## 4.2. Zero-shot Capabilities on Unseen Vision Tasks

**Settings**   To directly assess the task-level zero-shot capabilities of the decoder-only autoregressive transformer fine-tuned on the constructed DECVT, we exclude certain tasks from the DECVT training data. **Specifically, we remove data corresponding to *Image Restoration*, *Depth Estimation*, *Surface Normal Estimation*, *Depth-to-Image*, *Surface Normal-to-Image*, *HED-to-Image* and *HED Boundary Detection* tasks from the "Terminological-based Vision Tasks" component.** For quick validation, we utilize a subset of the DECVT training data: 30% data from the "Explanatory-based Vision Tasks" component (50% image editing data while the remaining 50% data from more visual-related image pairs) and 20% data from the remaining portion of the "Terminological-based Vision Tasks" component, with each subset containing approximately 0.5 million bidirectional pair of "image ↔ explanatory instruction ↔ image" triplets (totally 1.5 million bidirectional pair of triplets). The pre-trained model used to initialize is "Lumina-mGPT-7B-768". Training for the 7B AR-based VLM model is conducted on 8 A100 GPUs with batch size set as 128 over 2 epochs (around 46k iterations), totaling 1,340 GPU hours. Image resolution equals $448 \times 448$.

**Task-level Zero-shot Capabilities**   We evaluate the fine-tuned model's task-level zero-shot generalization capabilities on three previously unseen terminological-based vision tasks, *i.e.*, *HED-to-Image*, *Canny-to-Image*, and *Depth-to-Image*, as directly testing on these unseen vision tasks pro-

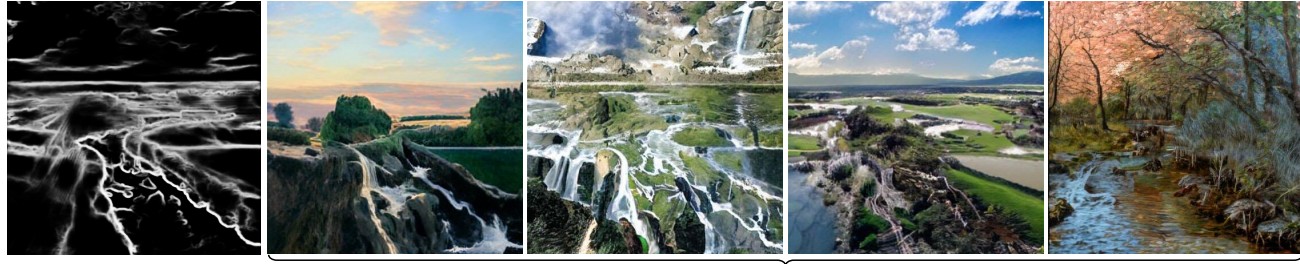

Input Image                                                    Output Image

*Figure 6.* Examples of task-level zero-shot capabilities (*HED-to-Image*). Resolution: 448×448. Explanatory Instruction: "`Gradually restore the scene's natural colors, filling in each region with realistic gradients and textures that represent natural elements. Reintroduce details such as color gradients in the sky, reflections in the water, and varied shades across fields and trees. Add depth by applying soft shading to convey lighting and shadow, ensuring the scene captures the natural flow of colors, reflections, and atmospheric lighting typical of a landscape under daylight.`"

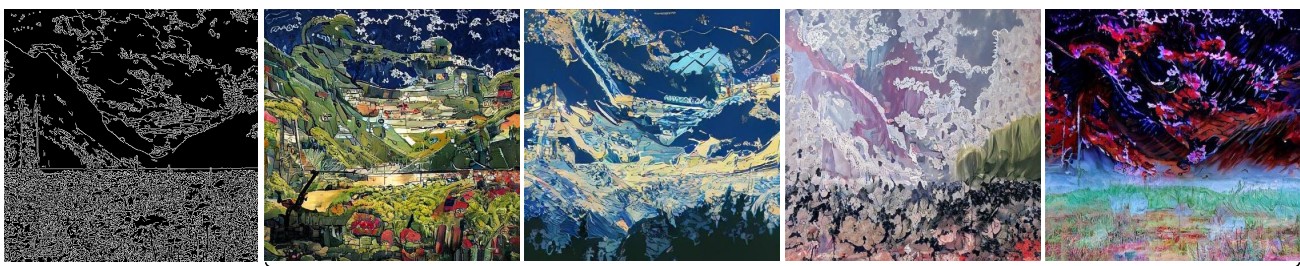

Input Image                                                    Output Image

*Figure 7.* Examples of task-level zero-shot capabilities (*Canny-to-Image*). Resolution: 448×448. Explanatory Instruction: "`Fill in all the empty outlines with rich colors that reflect vibrant tones, while redefining the shapes with smooth textures. Add layers of depth to the flat contours by enhancing brightness gradients in the sky, shadowing in the mountains, and intricate shades among the flowers. Reintroduce the sensation of open space and dimension by contrasting sharp objects with muted backgrounds and crisp details in the foreground.`"

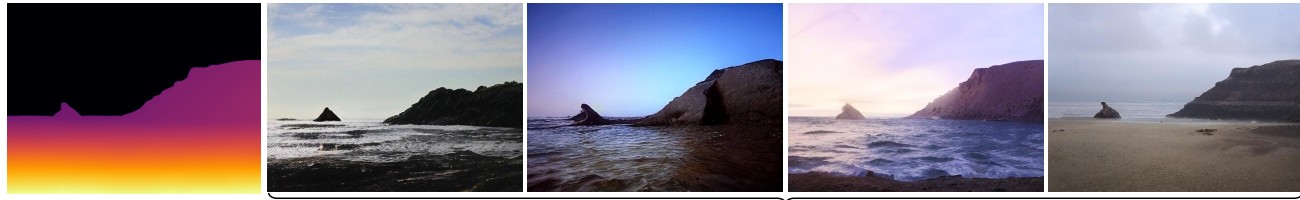

Input Image                                                    Output Image

*Figure 8.* Examples of task-level zero-shot capabilities (*Depth-to-Image*). Resolution: 448×448. Explanatory Instruction: "`Create a realistic coastal scene based on the image. Begin by interpreting the gradient colors to identify the relative distance of landscape elements. Use this information to construct a rocky coastline with a prominent headland and isolated rock. Add natural colors for sea and sky, and introduce texture and shading to the rocky surfaces, aiming to replicate a dusk setting with soft lighting and gentle waves.`"

vides the most intuitive demonstration of the model's task-level zero-shot potential. For each task, we provided linguistic descriptions, termed as "Explanatory Instructions", which detailed the desired transformations for the input images. These instructions, input along with the source images, guided the model in generating outputs that closely aligned with the specified transformations, as shown in Fig. 6, Fig. 7 and Fig. 8. These successful interpretation and generation processes, without prior exposure to these tasks, highlights the potential and feasibility of the proposed explanatory in-

structions in promoting task-level zero-shot learning within the vision domain. For additional task-level zero-shot examples, please refer to Appendix C.

### 4.3. Quantitative Results

In this section, we evaluate the fine-tuned AR-based VLM on generation tasks including *Canny-to-Image*, *HED-to-Image*, *Inpainting* and *Outpainting*, dense image prediction tasks including *Semantic Segmentation*, *Depth Estimation*

*Table 1.* Comparisons with task-specific / vision generalist baselines across four representative tasks. Our model adopts unseen explanatory instructions during the inference stage. We use GPT-4o (OpenAI, 2024) to recaption images in the MultiGen-20M (Qin et al., 2023) validation set as the original dataset do not provide captions in a general nature language format. Scores in the format of "a / b" indicate CLIP-Score for "previous / recaption" images and texts. "×" indicates that the method is incapable of performing the task. "-" indicates that the method does not report the corresponding results. "T. Z.-s." denotes task-level zero-shot, *i.e.*, both the task and instructions are not seen during training. "I. Z.-s." denotes instruction-level zero-shot, *i.e.*, the instructions are not seen during training.

| Methods | [T. Z.-s.] Canny-to-Image | | | [I. Z.-s.] HED-to-Image | | | [T. Z.-s.] Inpainting | | [T. Z.-s.] Outpainting | |
| --- | --- | --- | --- | --- | --- | --- | --- | --- | --- | --- |
| | F1↑ | FID↓ | CLIP-S↑ | SSIM↑ | FID↓ | CLIP-S↑ | FID↓ | LPIPS↓ | FID↓ | IS↑ |
| | MultiGen-20M | | | MultiGen-20M | | | Places | | Places | |
| Original Images | 100.00 | 0.00 | 31.99/33.13 | 1.0000 | 0.00 | 31.99/33.13 | 0.00 | 0.00 | 0.00 | 27.70 |
| ControlNet-SD1.5 (2023b) | 34.65 | **14.73** | **32.15** | 0.7621 | 15.41 | 32.33 | × | × | × | × |
| T2I-Adapter-SD1.5 (2024) | 23.65 | 15.96 | 31.71 | - | - | - | × | × | × | × |
| LDM-4 (2022) | × | × | × | × | × | × | 9.39 | 0.25 | - | - |
| LaMa (2022) | × | × | × | × | × | × | 12.00 | **0.24** | - | - |
| DeepFill v2 (2019) | × | × | × | × | × | × | - | - | 11.51 | 17.70 |
| MaskGIT (2022) | × | × | × | × | × | × | - | - | 7.80 | **22.95** |
| OmniGen (2024) | **35.54** | - | - | **0.8237** | - | - | - | - | - | - |
| PixWizard (2024) | 35.46 | 15.76 | 32.01 | - | - | - | **9.27** | 0.25 | **7.54** | 22.18 |
| Lumina-mGPT (2024) | 10.09 | 61.65 | 25.33 | 0.1834 | 69.49 | 25.97 | 42.69 | 0.77 | 42.66 | 11.28 |
| Ours | 20.69 | 26.93 | 27.16 | 0.6561 | 15.91 | 29.59 | 15.16 | 0.48 | 15.15 | 17.20 |

*Table 2.* Comparison with task-specific and vision generalist baselines across seven dense image prediction or low-level tasks. Our model adopts unseen explanatory instructions during the inference stage. "*" indicates the result is higher than typically expected under standard evaluation method and is provided for reference only. "×" indicates that the method is incapable of performing the task. "-" indicates that the method does not report the corresponding results. "T. Z.-s." denotes task-level zero-shot, *i.e.*, both the task and instructions are not seen during training. "I. Z.-s." denotes instruction-level zero-shot, *i.e.*, the instructions are not seen during training.

| Methods | [I. Z.-s.] Depth Est. | Semantic Seg.* | [I. Z.-s.] Surf. Norm. Est. | [I. Z.-s.] Derain | | [I. Z.-s.] Dehazing | | [T. Z.-s.] (UDC)IR | | [T. Z.-s.] Low-light Enh. | |
| --- | --- | --- | --- | --- | --- | --- | --- | --- | --- | --- | --- |
| | RMSE↓ | mIoU↑ | Mean Angle Error↓ | PSNR↑ | SSIM↑ | PSNR↑ | SSIM↑ | PSNR↑ | SSIM↑ | PSNR↑ | SSIM↑ |
| | NYU-Depth V2 | ADE20K | NYU-Depth V2 | Rain100L | | SOTS | | UDC(T-OLED) | | LOLv2 | |
| DepthAnything (2024) | **0.206** | × | × | × | × | × | × | × | × | × | × |
| Marigold (2024) | 0.224 | × | × | × | × | × | × | × | × | × | × |
| Mask DINO (2023a) | × | **60.80** | × | × | × | × | × | × | × | × | × |
| Mask2Former (2022) | × | 56.10 | × | × | × | × | × | × | × | × | × |
| Bae et al. (2021) | × | × | **14.90** | × | × | × | × | × | × | × | × |
| InvPT (2022) | × | × | 19.04 | × | × | × | × | × | × | × | × |
| AirNet (2022) | × | × | × | 32.98 | 0.951 | 21.04 | 0.884 | 26.76 | 0.799 | 19.69 | 0.821 |
| PromptIR (2023) | × | × | × | **36.37** | **0.972** | **30.58** | **0.974** | - | - | **21.23** | **0.860** |
| Unified-IO (2022) | 0.387 | 25.71 | - | × | × | × | × | × | × | × | × |
| Painter (2023) | 0.288 | 49.90 | × | 29.87 | 0.882 | - | - | - | - | - | - |
| InstructCV (2024) | 0.297 | 47.23 | × | × | × | × | × | × | × | × | × |
| InstructDiffusion (2024) | × | × | × | 19.82 | 0.741 | - | - | - | - | - | - |
| PixWizard (2024) | 0.287 | 36.76 | 19.65 | 31.43 | 0.917 | 28.14 | 0.937 | **27.22** | **0.826** | 20.29 | 0.807 |
| Lumina-mGPT (2024) | × | - | 28.49 | × | × | × | × | × | × | × | × |
| Ours | 0.553 | 42.12 | 27.21 | 16.72 | 0.462 | 16.90 | 0.542 | 14.54 | 0.332 | 14.48 | 0.371 |

and *Surface Normal Estimation*, low-level tasks including *Deraining*, *Dehazing*, *Image Restoration* and *Low-light Enhancement*. Training settings follow Sec. 4.1 and results are shown in Table 1 and Table 2. Details for these datasets, evaluation settings and more quantitative results please refer to Appendix B.1.2 and B.1.3.

# 5. Limitations and Discussions

In this section, we primarily focus on the most critical aspect—task-level zero-shot capabilities. For a more comprehensive discussion on limitations and additional insights (*e.g.*, dataset constraints and model stability), please refer to Appendix D. While the experiments in Sec. 4.2 and Appendix C demonstrate that models fine-tuned on datasets constructed with the proposed explanatory instructions exhibit task-level zero-shot capabilities, it is important to note that these capabilities are observed primarily in generation tasks (*e.g.*, *HED-to-Image*, *Canny-to-Image*, *Depth-to-Image*, *Outpainting* and *Inpainting*) and low-level vision tasks (*e.g.*, *Low-light Enhancement* and *Deblurring*). However, it fails to exhibit task-level zero-shot capabilities in vision tasks like *Image-to-Canny* or *Image-to-Depth*. Despite these limitations, it is important to highlight that the task-level zero-shot capabilities we observe are fundamentally different from the fixed-instruction forms commonly used in controllable generation tasks. In fixed-instruction scenarios, models not only know the task objectives but are also given the captions for the images to be generated. In contrast, by providing varied explanatory instructions, we

guide the model to understand the specific task objective, thereby expanding its ability to handle a diverse range of tasks.

Regarding the scope of task-level zero-shot capabilities, we hypothesize that the primary reason for this limitation is the lack of alignment between the image tokenizer (*e.g.*, VQ-VAE or VQ-GAN) and the text modality during pre-training. If the pretrained model used for initialization lacks the ability to generate images resembling Canny edges or depth maps, the fine-tuned model struggles to generalize to these tasks based solely on linguistic descriptions. In contrast, for controllable generation tasks such as *Canny-to-Image*, the text concepts (*e.g.*, "sky") have already been aligned with image tokens in the pre-trained model through paired text-image data. As a result, it is less affected by the lack of alignment between the image tokenizer and text modality, enabling task-level zero-shot capabilities through our proposed explanatory instructions.

## 6. Related Work

**Vision Task Understanding**   With the groundbreaking success of LLMs (Brown et al., 2020; Touvron et al., 2023a) in the field of NLP which have unified language tasks, researchers are no longer satisfied with earlier proprietary vision models (where each model handles a specific vision task). Instead, they seek to leverage the language modality to better understand the vision modality (Lian et al., 2025; Fu et al., 2025) and associated vision tasks. Advancements in these VLMs (Liu et al., 2023a; Li et al., 2023b; Alayrac et al., 2022; Dai et al., 2023; Zhu et al., 2023; Chen et al., 2024) have enabled substantial progress on vision-language tasks, achieving outstanding performance across diverse applications. However, early VLMs primarily produce textual outputs, which constrains their capacity to represent and manipulate visual information. To push beyond these limitations, researchers have explored integrating task-specific extensions such as expert models and downstream tools to enhance VLMs' ability to handle diverse downstream vision tasks (Liu et al., 2023c; Wu et al., 2023; Liu et al., 2023b; Fei et al., 2024; Huang et al., 2024; Gan et al., 2024; Geng et al., 2024; Wu et al., 2024). However, such simple extensions can not enable models to generalize across computer vision tasks. As the pursuit of Artificial General Intelligence (AGI) continues to grow, there is an increasing demand for unified foundational models that can tackle diverse vision tasks and achieve zero-shot generalization at vision task level.

**Vision Task Generalization**   Although early VLMs have demonstrated some degree of generalization ability, it largely limited to visual question answering, where the question answering capability inherently possessed by LLMs.

In the ongoing exploration to find a unified approach for handling diverse vision tasks and enabling vision task-level generalization, Chameleon (Team, 2024) is among the first to apply the same transformer architecture to sequences of both image and text tokens, enabling the generation of both text descriptions and image representations within the LLM paradigm. Show-O (Xie et al., 2024) also demonstrate unified models by combining text and image generation capabilities, while these models are limited to specific generation tasks. Lumina-mGPT (Liu et al., 2024), Omni-Gen (Xiao et al., 2024), and PixWizard (Lin et al., 2024) then leverage fixed or open-style terminological instructions, employing either LLMs or flow-based Diffusion Transformers (DiT) (Ma et al., 2024) as foundational architectures to integrate a wide range of vision tasks. Despite these advancements, these so-called vision generalist models still fail to exhibit vision task-level generalization. We infer the primary reason is that these models still rely on terminological instructions such as "Semantic segmentation", which confine the model to perform only the specific tasks defined by these terms or closely related tasks (*e.g.*, different forms of *Image Restoration*). In addition, the scope of these terminological instructions remain constrained and do not explain the true objective of vision tasks, far from the flexibility and richness of task expressions and objectives in the field of NLP. In this work, we move beyond the traditional notion of computer vision tasks and introduce explanatory instructions to explicitly articulate the objectives underlying different visual changes, forming an expansive set of unconstrained vision tasks that enable the model to achieve vision task-level generalization.

## 7. Conclusion

A significant body of research (Brown et al., 2020; Touvron et al., 2023a; Radford et al., 2021; Wei et al., 2023; Zhang et al., 2025) has demonstrated that, with sufficient training data, models can exhibit zero-shot generalization capabilities. Building on these findings, this work goes beyond the traditional notion of "vision tasks" by introducing explanatory instructions to reveal the true objectives underlying various vision tasks. In constructing the Dataset of Explanatory CV Tasks, we effectively captured and described a diverse range of non-terminological vision tasks. Through straightforward fine-tuning experiments on a vanilla token-based VLM, we showed that this approach enables the model to achieve both instruction-level and vision task-level zero-shot capabilities. Although the model still faces certain limitations, we believe this work represents a step toward achieving broader zero-shot generalization in computer vision by leveraging explanatory instructions. This method overcomes the constraints of conventional vision task definitions and paves the way for more flexible and universally applicable unified generative models.

## Acknowledgements

The authors would like to extend their sincere thanks to Xuhao Sun for his valuable discussions during the early stages of this work. Special thanks are also due to Mengxi Zhang and Chuyang Zhao for their contributions in reviewing and verifying the code. Additionally, we express our gratitude to the Chameleon team and the Lumina-mGPT team for their contributions to the open-source community.

## Impact Statement

This paper presents work whose goal is to advance the field of Machine Learning. There are many potential societal consequences of our work, none which we feel must be specifically highlighted here.

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

# A. Details for the Dataset of Explanatory CV Tasks

## A.1. Describing or Generating Explanatory Instructions

For certain terminological-based vision tasks (*e.g.*, Detection and Segmentation), we selected a subset of data and manually crafted descriptions based on the category and other label information. For other data, explanatory instructions were generated by constructing prompts and leveraging GPT-4o (OpenAI, 2024) for automated description generation.

For the manually crafted descriptions, we utilized human-written expressions alongside GPT-4o-generated outputs, applying rule-based combinations to produce over 5,000 unique forms of expression. Specifically, the entire *Object Detection* data, as well as the majority of data for *Segmentation*, *HED Boundary to Image*, *Surface Normal Estimation*, *Depth Estimation*, and their inverse tasks, were constructed through this approach.

For the GPT-4o-generated descriptions, early experiments utilized "gpt-4o-2024-05-13" to construct explanatory instructions. Later, for simpler sections of terminological-based vision tasks (*e.g.*, *HED Boundary to Image*, *Deraining*, *Dehazing*, *Desnowing*), we employed "gpt-4o-2024-08-06". For all other tasks, we used "chatgpt-4o-latest" to generate the instructions. The prompt design is summarized as follows (contents within "{**}" and "{==}" are optional while content in "{**}" is activated only when human evaluation deems GPT-4o's descriptions inaccurate):

```
1  response = client.chat.completions.create(
2      model = "chatgpt-4o-latest",#gpt-4o-2024-05-13, gpt-4o-2024-08-06
3      response_format = {"type": "json_object"},
4      messages = [
5          {
6              "role": "system",
7              "content": "You are an expert in computer vision and describe vison
                  tasks, with exceptional attention to detail. Your task is to
                  provide detailed descriptions of the transformations between the
                  images, explaining how elements appear, disappear, or change.
                  Your analysis is thorough, accurate, and insightful."},
8          {
9              "role": "user",
10             "content": [
11                 {
12                     "type": "text",
13                     "text": "{*We are currently working on tasks related to ...*}
                          Define the first image as A, the second image as B. Task
                          : {=1) Describe these images.=} 2) Describe 2 scenarios:
                          how to transform image A into image B, image B into image
                           A without referencing the contents of the other image
                          directly. Output format: JSON format with keys only
                          contain {=Image_A_Caption, Image_B_Caption,=}
                          Task_Descriptions_from_A_to_B,
                          Task_Descriptions_from_B_to_A without any nested JSON
                          structures. The descriptions of transformations should be
                           as diverse as possible, either as a single paragraph or
                          as a step-by-step description. Constraints: 1) The task
                          descriptions should not use any additional tools or
                          references, such as image editing tools. 2) Do not use
                          terms like 'image A', 'image B', 'to transform * into *',
                           or similar phrases."
14                 },
15                 {
16                     "type": "image_url",
17                     "image_url": {
18                         "url": f"data:image/jpeg;base64,{base64_image_A}",
19                         "detail": "high"
```

```
20                        }
21                    },
22                    {
23                        "type": "image_url",
24                        "image_url": {
25                            "url": f"data:image/jpeg;base64,{base64_image_B}",
26                            "detail": "high"
27                        }
28                    }
29                ]
30            }
31        ],
32        max_tokens = 4096
33 )
```

In addition, we experimented with several other models (*e.g.*, InternVL2-8B∼76B (Chen et al., 2024; OpenGVLab, 2024), QWen2.5 (Qwen Team, 2024), Claude-3.5-Sonnet (Anthropic, 2024), *etc.*) to generate explanatory instructions. However, to minimize potential biases in the generated explanatory instructions that could affect validation experiments, we ultimately selected GPT-4o. A comparison of selected models using the first image from the ADE20K Validation Set (Zhou et al., 2017b) is provided below (Table 3 ∼ Table 6). Content that does not meet the description requirements is highlighted in red.

**Caption of image A:**
1. Traditional stone building with a steeply pitched red-tiled roof.
2. Multiple chimneys and dormer windows.
3. Large white-framed windows.
4. Well-kept lawn in front.
5. Stone wall in front of the building.
6. Clear blue sky indicating a sunny day.

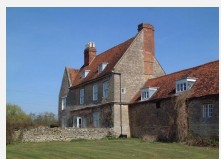

**Caption of image B:**
1. The same building with specific modifications.
2. Building and extension shaded with a semi-transparent reddish hue.
3. Sky colored a bright blue.
4. Lawn colored a vivid green.
5. Stone wall highlighted with a grayish color.
6. Bush to the right of the stone wall colored yellow.

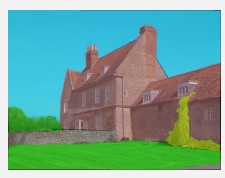

**Explanatory instruction from A to B:**
1. Color the Sky: 1) Select the sky area. 2) Fill the selected area with a bright, distinct color to make the sky stand out.
2. Shade the Building: 1) Select the entire building and its extension, including the roof and chimneys. 2) Apply a semi-transparent hue over the selected area to give it a shaded appearance while still allowing the underlying details to be visible.
3. Color the Lawn: 1) Select the lawn area in front of the building. 2) Fill this area with a bright, vibrant color to emphasize the grassy area.
4. Highlight the Stone Wall: 1) Select the stone wall in front of the building. 2) Apply a distinct, muted color to the wall to make it stand out from the lawn and building.
5. Color the Bush: 1) Identify and select the bush or shrub to the right of the stone wall. 2) Fill this area with a bright, distinct color to differentiate it from the surrounding elements.

**Explanatory instruction from B to A:**
1. Restore the Sky: 1)Select the area of the sky that has been filled with a bright, distinct color. 2) Replace the bright color with a natural sky color, making it appear more realistic.
2. Remove the Building Shading: 1) Select the area of the building and its extension shaded with a semi-transparent hue. 2) Remove the shading to restore the building's natural appearance, showing its original material and color.
3. Restore the Lawn Color: 1) Select the area of the lawn that has been filled with a bright, vibrant color. 2) Replace the bright color with a natural grass color to make it look realistic.
4. Remove the Highlight from the Stone Wall: 1) Select the stone wall that has been highlighted with a distinct, muted color. 2) Remove the highlighting to return the stone wall to its natural appearance.
5. Restore the Bush Color: 1) Select the bush or shrub to the right of the stone wall that has been colored with a bright, distinct color. 2) Replace the bright color with a natural color to make it look realistic and blend in with the surrounding environment.

*Table 3.* Examples of outputs from "gpt-4o-2024-05-13".

**Caption of image A:**
The image shows a large, two-story stone house with a red-tiled roof. The house has multiple chimneys and windows, some of which are dormer windows protruding from the roof. The building is surrounded by a well-maintained green lawn, and there is a stone wall in the foreground. The sky is clear and blue.

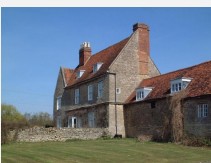

**Caption of image B:**
The image shows a large, two-story stone house with a red-tiled roof. The house has multiple chimneys and windows, some of which are dormer windows protruding from the roof. The building is surrounded by a well-maintained green lawn, and there is a stone wall in the foreground. The sky is clear and blue, but there is a noticeable green overlay on the right side of the house and lawn.

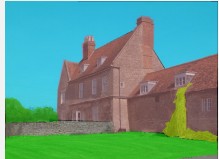

**Explanatory instruction from A to B:**
1: Change the sky color to a solid bright blue.
2: Add a green overlay to the grass area.
3: Add a green overlay to the ivy on the right side of the building.
4: Add a green overlay to the bushes on the left side of the building.
**Explanatory instruction from B to A:**
1: Change the sky color back to a natural blue with some cloud details.
2: Remove the green overlay from the grass area.
3: Remove the green overlay from the ivy on the right side of the building.
4: Remove the green overlay from the bushes on the left side of the building.

*Table 4.* Examples of outputs from InternVL2-76B.

**Caption of image A:**
This image shows a large stone farmhouse or manor house in a rural setting. The building has multiple stories with dormer windows on the roof. It features brick chimneys and a mix of stone and brick construction. There's a low stone wall in the foreground, and a well-maintained lawn area. The sky is clear and blue,indicating a sunny day. Some trees or vegetation can be seen in the background.

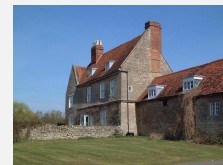

**Caption of image B:**
This image appears to be a digitally altered version of Image A. The basic structure and layout of the building remain the same, but the colors have been dramatically changed. The sky is now a vibrant, unnatural turquoise blue. The grass in the foreground is a bright, artificial-looking green. There's a yellow-green patch on the right side of the building that looks like it could be meant to represent ivy or some otherclimbing plant, but in a highly stylized way. The overall effect is surreal and cartoon-like.

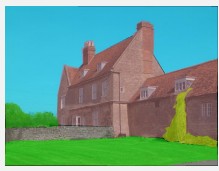

**Explanatory instruction from A to B:**
To transform Image A into something resembling Image B without referencing B's content:
1. Increase the color saturation of the entire image dramatically.
2. Change the sky to a bright, unnatural shade of blue.
3. Alter the grass color to a vivid, artificial-looking green.
4. Add a bright, contrasting color to one side of the building, perhaps representing vegetation in an exaggerated way.
5. Simplify the textures and details to create a more flat, cartoon-like appearance.
**Explanatory instruction from B to A:**
To transform Image B into something resembling Image A without referencing A's content:
1. Reduce the overall color saturation of the image significantly.
2. Adjust the sky color to a more natural, pale blue tone.
3. Change the grass to a more muted, realistic shade of green.
4. Remove any unnaturally bright or contrasting colors from the building and surroundings.
5. Enhance the texture and details of the building to create a more realistic, photographic appearance.

*Table 5.* Examples of outputs from "claude-3-5-sonnet-20240620".

**Caption of image A:**

The image captures a serene scene of a traditional English country house, nestled amidst a lush green lawn. The house, painted in a pristine white, stands out against the clear blue sky. It features a prominent red brick chimney, adding a touch of rustic charm to its appearance.

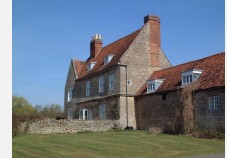

The house is adorned with multiple windows, each framed by white shutters, allowing ample natural light to filter into the rooms within. A large wooden door, painted in a contrasting dark brown color, serves as the main entrance to the house.

A stone wall encircles the property, providing privacy and security. The wall is punctuated by a gate, inviting visitors into the tranquil haven beyond. The house is surrounded by an expanse of well-maintained grass, creating a sense of openness and tranquility.

Despite the detailed description, there's no text visible in the image, keeping the focus solely on the visual elements. The relative positions of these objects suggest a well-planned layout, with the house at the center, the stone wall forming a boundary around it, and the gate marking the entry point.

Overall, the image presents a picturesque view of a typical English countryside setting, where nature and architecture coexist harmoniously.

**Caption of image B:**

This image showcases a traditional English country house, set against a backdrop of a clear blue sky. The house,constructed from red brick, boasts a prominent chimney and a large wooden door. The roof, coveredin tiles, adds a touch of rustic charm to the overall structure.

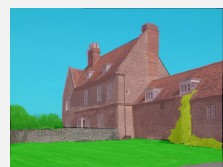

The house is situated on a lush green lawn, which is enclosed by a stone wall. This wall not only provides a boundary but also enhances the aesthetic appeal of the property. A gate can be seen in the wall, possibly serving as the entrance to the property.

In terms of object count, there's one house, one chimney, one door, and one gate. The relative position of these objects suggests a well-planned layout - the house is centrally located, the chimney is attached to the house, the door is part of the house, and the gate is integrated into the stone wall surrounding the property.

There's no text present in the image, keeping the focus solely on the visual elements. The image does not provide any information about the actions or movements of the objects, as they are stationary structures.

Overall, the image presents a picturesque view of a typical English countryside setting, where nature and architecture coexist harmoniously.

**Explanatory instruction from A to B:**

Adjust Color Scheme: Begin by altering the color palette of the main building. You could introduce warmer hues to the exterior, perhaps changing the white walls to a soft cream or adding accents of natural stone colors to give it a different character.

Modify Architectural Features:Modify architectural elements such as the chimney and windows. For instance, you might extend the chimney height, add decorative brickwork, or replace the window shutters with functional window boxes filled with blooming flowers.

Landscape Transformation: Redesign the surroundings by introducing a garden path leading up to the house or adding a small pond near the entrance. Change the lawn to incorporate more diverse flora, like flower beds or shrubberies.

Add or Remove Structures: Introduce a detached garage or a garden gazebo in the background, or conversely, remove existing elements like the stone wall to open up the view and create a more expansive landscape.

Lighting and Atmospheric Effects: Adjust the lighting to simulate a different time of day, such as sunset or twilight, and add atmospheric effects like mist or rain to alter the mood of the scene.

**Explanatory instruction from B to A:**

Reverse Color Alterations: Restore the original color palette of the building, possibly reverting any warm tones added in the previous transformation back to cooler shades, like the classic white or a more subtle hue that contrasts with the greenery.

Revert Architectural Modifications: Remove any additional architectural features introduced, such as decorative chimney work or new window features, restoring the building to its simpler, more traditional form.

Revive Original Landscape: Return the landscaping to its initial simplicity, eradicating any new paths, ponds, or additional structures like gazebos that were added. Reinstate the uniform green lawn and, if removed, rebuild the enclosing stone wall.

Alter Atmospheric Conditions: Adjust the lighting and weather conditions to their prior state, perhaps brightening the scene to reflect midday sunlight or clearing away any atmospheric effects like fog or rain.

Restore Original Scene Balance: Ensure all changes collectively bring the scene back to a state reminiscent of the original, while not precisely replicating Image A, focusing on balancing the composition and elements to achieve a similar ambiance.

*Table 6.* Examples of outputs from Qwen2.5.

## A.2. Generating Data Samples

For a subset of terminological-based vision tasks, we employed existing methods to generate image pairs.

**HED Boundary Detection & HED Boundary to Image.** We use HED boundary detection to generate approximately 10k edge-image pairs from randomly selected images in other datasets.

**Image Depth Estimation & Depth Map to Image.** We used the Depth Anything V2 model (Yang et al., 2024) and obtain about 250k depth-image pairs. Original images are randomly selected from other datasets.

**Surface Normal Estimation & Normal Map to Image.** We generated the surface normal maps by using DSINE (Bae & Davison, 2024). We obtain around 250k normal-image pairs. Original images are randomly selected from other datasets.

## A.3. More Data Samples for the Dataset of Explanatory CV Tasks

In this section, we provide more data samples for the Dataset of Explanatory CV Tasks.

### A.3.1. MORE DATA SAMPLES FOR EXPLANATORY-BASED VISION TASKS

More data samples for "Explanatory-based Vision Tasks" are provided in Fig. 9 ∼ 16.

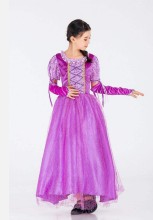

**Explanatory instruction from A to B:**
The dress changes from having long sleeves with ribbons to having off-the-shoulder sleeves. The color of the dress shifts from deep purple to a lighter shade with a satin finish. The hairstyle transitions from braided and tied back to hair let down with a tiara. Gloved hands shift to being ungloved with a visible change in the hand accessory. Socks become part of the footwear, altering from lace-up shoes to white socks. The expression shifts from side-facing to a front-facing stance with a visible smile.

**Explanatory instruction from B to A:**
Adjust the dress design from off-the-shoulder with satin to one featuring long sleeves with ribbon details. Modify the dress color from a light, shiny appearance to a deep purple tone with a more textured fabric. Change the hairstyle from loose hair with a tiara to a braided style pulled back. Introduce gloves on the hands, replacing the bare hand appearance. Transition the socks to dark shoes, altering the footwear style. Shift the facial expression from a smiling, forward-facing look to a neutral, side-facing demeanor.

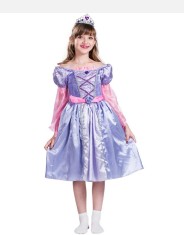

*Figure 9.* Data sample for explanatory-based vision tasks.

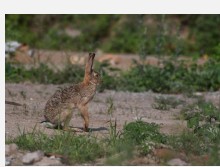

**Explanatory instruction from A to B:**
Shake up the perspective to bring the animal slightly closer, reposition upright with a slight turn of the head. Reduction of debris gives the ground a smoother, even appearance.

**Explanatory instruction from B to A:**
Adjust the animal's posture to a more upright sitting position, shifting the angle to reveal more surroundings. Enhance the scene with additional greenery and scattered natural elements.

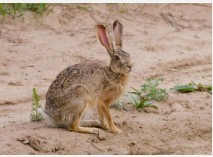

*Figure 10.* Data sample for explanatory-based vision tasks.

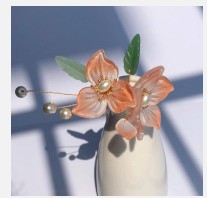

**Explanatory instruction from A to B:**
Reposition the vase so that it is more upright, causing the flowers to shift slightly to the left. Adjust the angle of the sunlight so that shadows fall more prominently to the bottom right, emphasizing a more elongated shadow of the floral arrangement. Rotate the floral elements along their central axis, making each petal and leaf face slightly different directions while maintaining their original colors and textures.

**Explanatory instruction from B to A:**
Tilt the vase towards the viewer, altering the flowers' positions to the right. Modify the lighting to change the direction of the shadows, creating a more compact and less defined shadow below the arrangement. Slightly adjust the orientation of the petals and leaves, ensuring they align in a different configuration yet retain their original forms and hues.

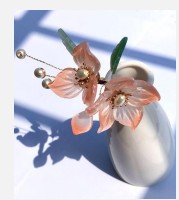

*Figure 11.* Data sample for explanatory-based vision tasks.

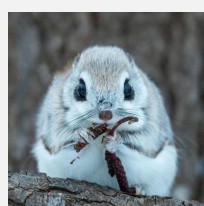

**Explanatory instruction from A to B:**
Brighten the overall scene, adding a more sunlit ambiance. Modify the gripping object to reflect a natural, green element while altering the environment to a more elevated, branch-like setting.

**Explanatory instruction from B to A:**
Soften the vivid warmth by introducing cooler tones. Exchange the greenery for a different object to hold, and replace the leafy background with a consistent, textured surface.

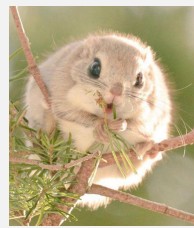

*Figure 12.* Data sample for explanatory-based vision tasks.

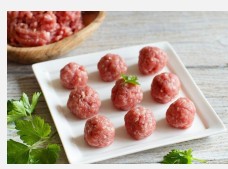

**Explanatory instruction from A to B:**
Replace the white square base with a large wooden board. Display the spheres in a less strict arrangement. Incorporate garlic bulbs and a scattering of spices in the background, complemented by soft cloth textures.

**Explanatory instruction from B to A:**
Shift the spheres onto a neat, white square template. Simplify the herb decoration to a single leaf per sphere and clean up the overall setting for a minimalist and fresh appearance.

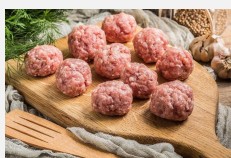

*Figure 13.* Data sample for explanatory-based vision tasks.

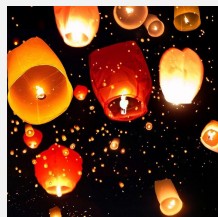

**Explanatory instruction from A to B:**
Add a crowd at the bottom with open hands raised towards the sky, and increase the number of lanterns, making them appear smaller as they move upwards. Darken the background slightly and include a distant bright light at the horizon.

**Explanatory instruction from B to A:**
Remove the crowd and their raised hands from the bottom, leaving only the lanterns. Brighten individual lanterns and make the scene more clustered with fewer visible lanterns.

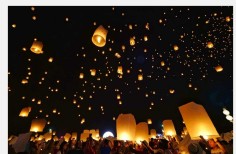

*Figure 14.* Data sample for explanatory-based vision tasks.

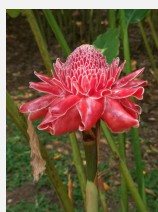

**Explanatory instruction from A to B:**
Introduce a background featuring a variety of green leaves and stems, while adding another flower similar to the first one. Ensure the new flower has a more fully opened appearance and a slightly different arrangement of petals.

**Explanatory instruction from B to A:**
Focus on a single flower, removing the secondary bloom and simplifying the background to emphasize the central floral subject. Strip away diverse leaves and stems, leaving a plain backdrop to highlight the singular floral element.

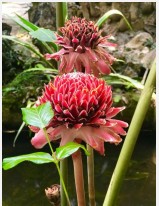

*Figure 15.* Data sample for explanatory-based vision tasks.



**Explanatory instruction from A to B:**
Adjust the color scheme from a blue-green light spectrum to a combination of red and green, creating a shift in the color dynamics. Alter the central projection, ensuring that the beam patterns reflect less linearity and more concentric circular formations on the surface. Intensify the red hues across the visible light paths and enhance the circular designs with a stronger emphasis on the differences in color, resulting in more elaborate geometric patterns.

**Explanatory instruction from B to A:**
Change the predominant color from red-green to a blue-green spectrum, modifying the overall illumination to convey cooler tones. Modify the light projections on the surface, transitioning from circular formations to a design where the beams are more linear and directed from the center point outward. Reduce the complexity of the geometric patterns, focusing on the linear symmetry, and diminish the red hues, emphasizing blues with a vibrant green accent.

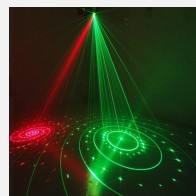

*Figure 16.* Data sample for explanatory-based vision tasks.

### A.3.2. MORE DATA SAMPLES FOR TERMINOLOGICAL-BASED VISION TASKS

More data samples for "Terminological-based Vision Tasks" are provided in Fig. 17 ∼ 31.

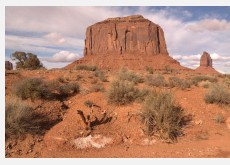

**Explanatory instruction from A to B:**
Enhance the overall saturation and vibrancy of the image by increasing color intensity, bringing out a more vivid blue sky and deepening the reds in the earth tones. Adjust the contrast to sharpen the details of the rocky formations.

**Explanatory instruction from B to A:**
Reduce the saturation and brightness to subdue the colors, reverting the image to a more natural state. Soften the contrast to give the image a more balanced and moderate appearance.

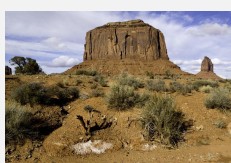

Figure 17. Data sample for terminological-based vision tasks (*Image Restoration*).

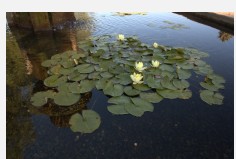

**Explanatory instruction from A to B:**
Enhance color saturation while maintaining the current brightness. Emphasize cooler tones for a more vibrant and lively appearance.

**Explanatory instruction from B to A:**
Reduce color saturation to achieve a more neutral tone. Neutralize cooler tones to recreate the initial natural look.

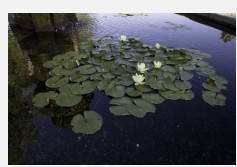

Figure 18. Data sample for terminological-based vision tasks (*Image Restoration*).

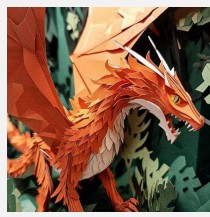

**Explanatory instruction from A to B:**
Drain the image of all bright colors, shifting the mood into a sepia-like monochrome. The dragon figure remains, but now weathered and eroded, appearing as if composed of twisted, dried-out textures. The background morphs dramatically as the lush flora converts into a barren, tangled form entirely overtaken by shadows and creeping vines. The resulting atmosphere becomes dark and decayed, contrasting the lively, colorful origin.

**Explanatory instruction from B to A:**
Reimagine the desolate, worn textures by replacing them with lively, bright tones. Breathe life and energy back into the figure by restoring its facial features, sharpening its horns, and returning its surroundings to a thriving, colorful forest, full of sharp contrasts and distinctly separated flora. Soften the twisted appearance of the dragon, allowing its scales to return to a more uniform, sleek form, where bold colors and vivid contrasts dominate the atmosphere.

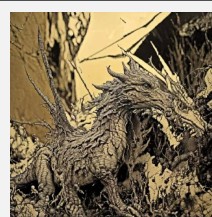

Figure 19. Data sample for terminological-based vision tasks (*Style Transfer*).

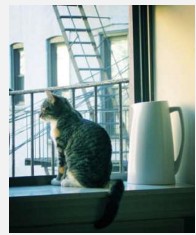

**Explanatory instruction from A to B:**
Maintain the layout but adjust the overall structure by shifting the visual style to a more artistic, painted form. The material should evolve into a representation with stronger outlines and exaggerated contrasts between shadows and highlights. All elements transition to bold blocks of color and simplified features, giving the appearance of something more abstract, yet still retaining the core elements of the original composition.

**Explanatory instruction from B to A:**
Soften all of the rough, exaggerated lines and deep hues present in the scene. Gradually smooth out all blocks of visual complexity forming natural color transitions between shadows and bright areas. Pull the bright elements into a more neutral, natural tone, eliminating hard contrasts. Return the visuals to a lifelike depiction, but keep the core layout intact.

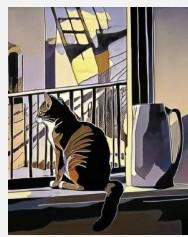

*Figure 20.* Data sample for terminological-based vision tasks (*Style Transfer*).

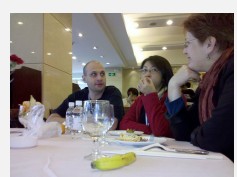

**Explanatory instruction from A to B:**
For the following object categories, apply the corresponding solid color overlays to fully cover them:
Color every wineglass object with a aqua solid layer.
Paint each fork with a linen solid color fill.
Color each banana object with a darkolivegreen solid overlay.
Paint each spectacles with a purple solid color fill.
Apply a solid lawngreen color overlay to fully cover all scarf objects.

**Explanatory instruction from B to A:**
Delete the solid color covering the objects, restoring their original look.

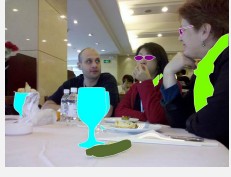

*Figure 21.* Data sample for terminological-based vision tasks (*Segmentation*).

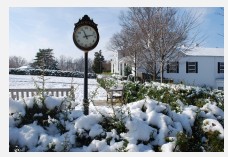

**Explanatory instruction from A to B:**
Paint over every clock, streetlight object with dodgerblue, covering them entirely.

**Explanatory instruction from B to A:**
Erase the dodgerblue overlay on the clock, streetlight objects, restoring the original appearance.

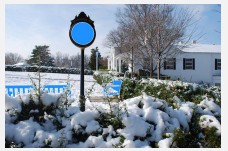

*Figure 22.* Data sample for terminological-based vision tasks (*Segmentation*).

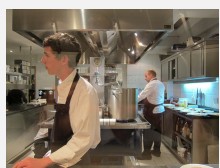

**Explanatory instruction from A to B:**
Draw a floralwhite bounding box around all apron objects.

**Explanatory instruction from B to A:**
Remove all bounding boxes from the objects, restoring the original image.

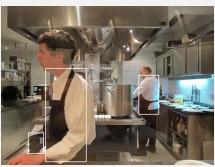

*Figure 23.* Data sample for terminological-based vision tasks (*Object Detection*).

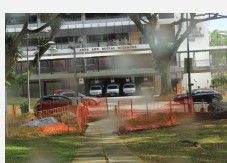

**Explanatory instruction from A to B:**
Remove the raindrop patterns that obscure sections of the scene to reveal a clear, unobstructed view. Enhance the visibility of details by eliminating blurred spots. Increase the sharpness and contrast to accentuate the overall clarity, ensuring all objects and surfaces are distinctly visible.

**Explanatory instruction from B to A:**
Introduce raindrop patterns on the lens to create areas of distortion and blur across the scene, simulating a rainy effect. Soften certain details by incorporating these smudged patterns, which partially obscure parts of the view. Decrease the contrast and sharpness to mimic the presence of rain on the camera lens.

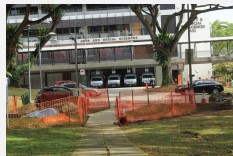

*Figure 24.* Data sample for terminological-based vision tasks (*Deraining*).

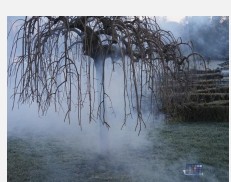

**Explanatory instruction from A to B:**
Introduce a layer of fog that softly envelops the scenery, generating a slightly diffused appearance with reduced contrast. Increase the density of mist, particularly around the lower parts of the tree and the ground, to create a moody and mysterious ambiance. Soften the outlines of distant elements and elements in the foreground, diminishing the clarity to simulate a hazy atmosphere. This results in a muted color palette, lending an ethereal quality to the overall visual experience.

**Explanatory instruction from B to A:**
Reduce the visibility of mist around the scene by gradually decreasing the density of fog, improving the clarity and sharpness of the landscape. Carefully remove any diffused light scattering, allowing the intricate details of the surrounding environment to emerge more distinctly. Ensure that the distant elements become more defined and that the colors appear more saturated and less muted, providing an unobstructed view of the complete scene.

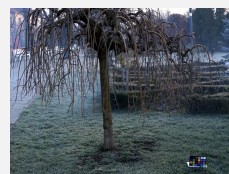

*Figure 25.* Data sample for terminological-based vision tasks (*Dehazing*).

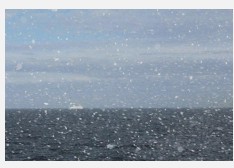

**Explanatory instruction from A to B:**
Reduce the visibility of falling particles obscuring the background by removing white speckles scattered across the scene. Clarify the view by eliminating opaque overlays, focusing on enhancing the details of the distant objects and sky. Enhance the overall brightness and contrast to highlight the serene atmosphere with unobstructed sightlines.

**Explanatory instruction from B to A:**
Introduce a layer of translucent white specks distributed unevenly across the scene to simulate a snowy condition. Gradually decrease the clarity of distant elements by overlaying semi-transparent textures while maintaining the overall composition. Weave a sense of dynamic motion through the addition of irregular shapes and lighter shades throughout the field, mimicking a flurry of snowfall.

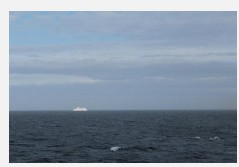

Figure 26. Data sample for terminological-based vision tasks (*Desnowing*).

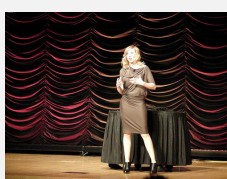

**Explanatory instruction from A to B:**
Simplify the scene into a grayscale silhouette, emphasizing the outlines and contours of prominent subjects while eliminating detailed features and textures. Fade out background patterns into a smooth gradient to suggest depth. Focus on transitioning the entire scene into varying shades of gray, concentrating on the key shapes and forms.

**Explanatory instruction from B to A:**
Introduce complex patterns and vibrant colors to the scene, adding texture and intricate details to key subjects. Enhance foreground subject clarity, incorporating color variations and shading. Overlay the background with rich details and vivid patterns to create a more detailed and nuanced appearance, emphasizing depth with contrasting hues.

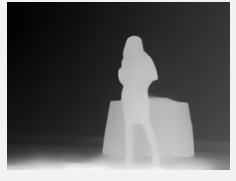

Figure 27. Data sample for terminological-based vision tasks (*Depth Estimation*).

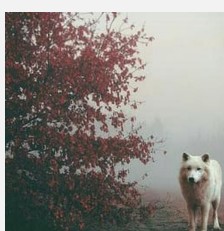

**Explanatory instruction from A to B:**
Introduce a series of vividly colored dots and lines across the canine figure. Each line should connect to form an intricate pattern that traces the outline and features of the animal, adding a whimsical and abstract design element.

**Explanatory instruction from B to A:**
Remove all the colored dots and lines overlaying the animal. Preserve the misty background along with the tree and the canine, ensuring the environment exudes quietude and mysticism.

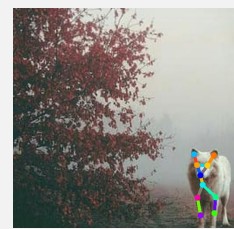

Figure 28. Data sample for terminological-based vision tasks (*Pose Estimation*).

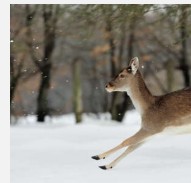

**Explanatory instruction from A to B:**
Focus solely on the anatomical framework by removing all environmental and bodily details, leaving only colorful joint lines on a dark canvas to abstractly outline movement.

**Explanatory instruction from B to A:**
Reintroduce the snowy environment, captured with a deer in motion, and omit the line structure to display a natural scene of winter and untouched wildlife activity.

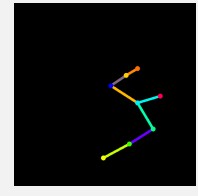

*Figure 29.* Data sample for terminological-based vision tasks (*Pose Estimation*).

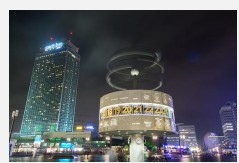

**Explanatory instruction from A to B:**
Apply a transformation that generalizes the depiction of depth and orientation using a colorful gradient representing angles across the surface. Structures become abstract with smooth transitions between colors, translating the detailed textural elements into flat gradients indicating spatial variations. The skyline should shift into simplified forms with no explicit textures, instead presenting undulating colors to denote geometric relationships. Emphasize prominent architectural features by focusing on their orientation, using color to suggest their form without explicit details. The luminance of the scene should be replaced with a vibrant, multi-hued representation that suggests the directionality and depth of each element.

**Explanatory instruction from B to A:**
Introduce detailed textural elements and a realistic color scheme over the abstract tonal surfaces, replacing the gradient with distinct textures that convey material properties and illumination. Reconstruct the background to include various light sources and urban details, accentuating the contrast and natural colors found on physical architectures. Re-establish the shadows and highlights to convey depth using realistic lighting that defines the shapes explicitly and clearly. Transition the simplified forms into detailed and distinct buildings, featuring precise windows, lighting, and signs that illustrate an urban setting at night, capturing the variations in illumination and environment.

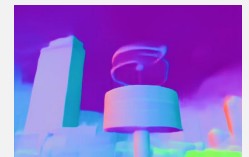

*Figure 30.* Data sample for terminological-based vision tasks (*Surface Normal Estimation*).

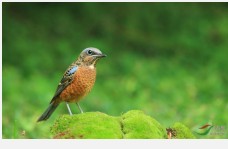

**Explanatory instruction from A to B:**
Highlight only major edges, transforming the image into a boundary map.

**Explanatory instruction from B to A:**
Convert boundary outlines into a realistic image, applying textures and colors.

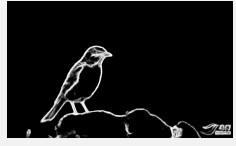

*Figure 31.* Data sample for terminological-based vision tasks (*HED Boundaries*).

# B. Results, Details and Samples for Quantitative Experiments

## B.1. Quantitative Experiments

Training settings follow Sec. 4.1 of the paper.

*Table 7.* Comparisons with image-editing baselines evaluated on the Emu Edit and MagicBrush test set. "-" indicates that the method does not report the corresponding results.

| Methods | Emu Edit test set | | | | MagicBrush test set | | | |
|---|---|---|---|---|---|---|---|---|
| | $CLIP_{im}\uparrow$ | $CLIP_{out}\uparrow$ | $\ell_1\downarrow$ | DINO$\uparrow$ | $CLIP_{im}\uparrow$ | $CLIP_{out}\uparrow$ | $\ell_1\downarrow$ | DINO$\uparrow$ |
| InstructPix2Pix (2023) | 0.834 | 0.219 | 0.121 | 0.762 | 0.837 | 0.245 | 0.093 | 0.767 |
| MagicBrush (2023a) | 0.838 | 0.222 | 0.100 | 0.776 | 0.883 | 0.261 | 0.058 | 0.871 |
| PnP (2023) | 0.521 | 0.089 | 0.304 | 0.153 | 0.568 | 0.101 | 0.289 | 0.220 |
| Null-Text Inv. (2023) | 0.761 | 0.236 | 0.075 | 0.678 | 0.752 | 0.263 | 0.077 | 0.664 |
| Emu Edit (2024) | **0.859** | 0.231 | 0.094 | **0.819** | **0.897** | 0.261 | **0.052** | **0.879** |
| UltraEdit (2024) | 0.844 | 0.283 | 0.071 | 0.793 | 0.868 | - | 0.088 | 0.792 |
| OmniGen (2024) | 0.836 | 0.233 | - | 0.804 | - | - | - | - |
| PixWizard (2024) | 0.845 | 0.248 | **0.069** | 0.798 | 0.884 | 0.265 | 0.063 | 0.876 |
| Lumina-mGPT (2024) | 0.815 | 0.283 | 0.149 | 0.735 | 0.855 | 0.282 | 0.114 | 0.786 |
| Ours | 0.821 | **0.286** | 0.132 | 0.768 | 0.875 | **0.292** | 0.093 | 0.831 |

### B.1.1. IMAGE EDITING RESULTS

We evaluate the fine-tuned AR-based VLM on two image editing benchmarks, *i.e.*, the MagicBrush test set (Zhang et al., 2023a) and the Emu Edit test set (Sheynin et al., 2024), which include several human-defined operations, *e.g.*, background alteration, object removal, object addition, localized modifications, *etc*. In alignment with the evaluation metrics used both by Emu Edit and MagicBrush, we measure four metrics: 1) $CLIP_{im}$: CLIP image similarity between source image and output image; 2) $CLIP_{out}$: CLIP text-image similarity between edited image and target caption; 3) $\ell_1$: $\ell_1$ distance between source image and output image; 4) DINO: DINO similarity between source image and output image.

As shown in Table 7, compared to instruction-guided image editing methods such as InstructPix2Pix (Brooks et al., 2023), PnP (Tumanyan et al., 2023), and Null-Text Inv.(Mokady et al., 2023), the straightforward fine-tuned AR-based VLM demonstrates certain advantages. However, when compared to more advanced methods such as Emu Edit(Sheynin et al., 2024), UltraEdit (Zhao et al., 2024), and recent VLMs employing flow-based Diffusion Transformers (DiT) (Ma et al., 2024) as backbones (*e.g.*, OmniGen (Xiao et al., 2024) and PixWizard (Lin et al., 2024)), the performance of our model still shows a noticeable gap. Nevertheless, compared to a vanilla token-based VLM (*i.e.*, Lumina-mGPT (Liu et al., 2024)), the fine-tuned model achieves significant performance improvements (Chameleon is excluded due to the lack of image generation capability).

### B.1.2. IMAGE GENERATION RESULTS

We then evaluate the effectiveness of generation capabilities. We assess its performance across three tasks: controllable image generation (including *Canny-to-Image* and *HED-to-Image*), *Inpainting*, and *Outpainting*. Results are shown in Table 1 in the main paper.

**Controllable Image Generation** We adopt the evaluation split of MultiGen-20M for *Canny-to-Image* and *HED-to-Image* condition. As MultiGen-20M (Qin et al., 2023) do not provide image captions in a general nature language format and the dataset is not used during training, we use GPT-4o (OpenAI, 2024) to recaption them. We also provide unseen explanatory instructions for each controllable generation data pair (the construction method for explanatory instructions can refer to Appendix A.1) to evaluate our model's instruction-level zero-shot capability. Following ControlNet++ (Li et al., 2024), we assess controllability by measuring the similarity between the input conditions and the extracted conditions from the generated images. We use F1-Score for *Canny-to-Image* and SSIM for *HED-to-Image*. We adopt FID and CLIP-Score to evaluate the quality of the generated images and their alignment with caption. The resolution of generated images is $448 \times 448$ but are resized to $512 \times 512$ for evaluation (output resolution for Lumina-mGPT is $512 \times 512$). Results

are shown in Table 1. As both *Canny Edge Detection* and *Canny-to-Image* tasks are not seen during training, results for *Canny-to-Image* represent the more challenging task-level zero-shot setting. Instructions for Lumina-mGPT used fixed format (Lumina-mGPT does not work if change the instruction format, details can refer to Appendix B.2). However, evaluate instructions for ours are unseen during training. While the results still fall short of those achieved by task-specific or so-called vision generalist models, it demonstrates significant progress compared to previous models, which were largely incapable of zero-shot generalization at either the instruction level or the task level.

**Inpainting and Outpainting** For image inpainting, we use random rectangle or circle to mask 40%-50% of the image area and measure FID and LPIPS to assess the quality of the generated images. For image extrapolation (outpainting), we follow MaskGIT (Chang et al., 2022) settings, *i.e.*, extending the image by 50% to the right and use FID and Inception Score (IS) to assess the quality of the generated images. We do not use image captions that directly describe the output images but construct unseen explanatory instructions during inference stage for our model (details can refer to Appendix A.1). Both the inpainting task and the outpainting task are not seen during training (but inpainting may have task-level overlap with *Segmentation-to-Image*). We evaluated on 10,000 image crops from the Places dataset (Zhou et al., 2017a) (training set for the Places dataset are not used during our training stage). The size of generated images is $448 \times 448$ but we resize images to $512 \times 512$ for evaluation (output resolution for Lumina-mGPT is $512 \times 512$). As shown in Table 1, although the results of our generated images still show a gap compared to other task-specific or vision generalist models, we represent a significant breakthrough in zero-shot capability.

### B.1.3. RESULTS FOR DENSE IMAGE PREDICTION AND LOW-LEVEL TASKS

We provide more details for dense image prediction tasks and low-level tasks in this section. Results are shown in Table 2 in the main paper.

**Dense Image Prediction** We conduct *Semantic Segmentation* experiments on the ADE20K validation set (Zhou et al., 2017b), while *Depth estimation* and *Surface Normal Estimation* experiments on the NYU-Depth V2 dataset (Silberman et al., 2012). For *Semantic Segmentation*, since we use color to control category generation and managing all categories simultaneously leads to higher color inaccuracies, we test each category individually during the evaluation stage. This approach yields results higher than conventional evaluation methods, and the result is provided as reference only. Accuracy is evaluated using the Mean Intersection over Union (mIoU) metric. For the monocular *Depth Estimation* task, we adjust the depth values of the generated image to fall within the range of zero to ten meters. Accuracy is measured using the Root Mean Square Error (RMSE). For the *Surface Normal Estimation* task, we recover the corresponding normal vectors from the output image and assess accuracy using the Mean Angle Error metric. For both *Depth Estimation* and *Surface Normal Estimation* tasks, we employ unseen explanatory instructions during evaluation to guide the model toward the target objective.

**Low-level Tasks** We conduct *Deraining* experiments on the Rain100L (Yang et al., 2017) dataset and *Dehazing* experiments on the SOTS (Li et al., 2018) dataset, both using unseen explanatory instructions during the evaluation stage. Subsequently, we perform task-level zero-shot evaluations on under-display camera (UDC) *Image Restoration* task using the UDC (T-OLED) dataset (Zhou et al., 2021) and on *Low-light Enhancement* task using the LOLv2 dataset (Yang et al., 2021). Similarly, the explanatory instructions used during evaluation are unseen during the training phase. We evaluate performance using PSNR and SSIM as distortion metrics.

Although the results of the generated images still show a gap compared to task-specific or existing vision generalist models, our model successfully understand the task objectives conveyed by the unseen explanatory instructions and effectively complete the corresponding tasks. Unseen explanatory instruction samples for these tasks can be found in Section B.2 (cf. Fig. 33 ∼ 43). While examples of the generated images for these tasks can be found in Section C.

### B.1.4. MORE MERITS OF EXPLANATORY INSTRUCTIONS

For vision tasks have already been included in the training set, explanatory instructions can also contribute to the understanding of zero-shot samples. As shown in Fig. 32, for categories present in the training set (*e.g.*, turtle), both direct labels like "turtle" and descriptive phrases such as "the creature swimming in the water" effectively guide the model in completing tasks. However, for categories not included in the training set (*e.g.*, broad-winged damselfly), the model struggles to interpret the category name alone but can be assisted by descriptive expressions like "the creature on the leaf".

**Explanatory Instruction**: "Apply a red color overlay to the turtle."

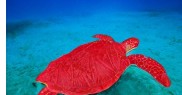

**Output Image**

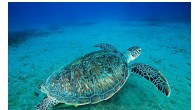

**Input Image**

**Explanatory Instruction**: "Apply a red color overlay to the creature swimming in the water in the image."

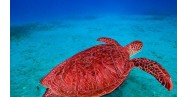

**Output Image**

**Explanatory Instruction**: "Apply a red color overlay to Broad-winged damselflies."

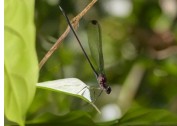

**Output Image**

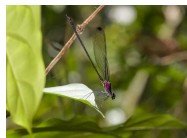

**Input Image**

**Explanatory Instruction**: "Apply a red overlay to the creature on the leaf in the image."

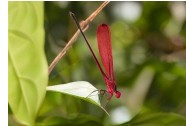

**Output Image**

**Explanatory Instruction**: " Change the color of aegopodium podagraria in the picture to purple."

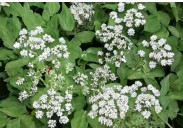

**Output Image**

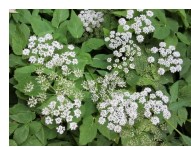

**Input Image**

**Explanatory Instruction**: " Change the color of the white flower-like plants in the picture to purple."

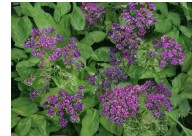

**Output Image**

**Explanatory Instruction**: " Cover elaeagnus umbellata with blue."

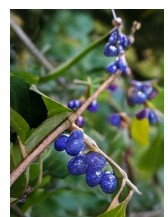

**Output Image**

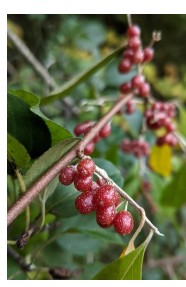

**Input Image**

**Explanatory Instruction**: " Cover the red fruit in the picture with blue."

**Output Image**

*Figure 32.* Explanatory instructions assist the model in understanding task objectives. (*Segmentation*).

## B.2. Details and Unseen Instruction Samples for Quantitative Experiments

In this section, we provide a more detailed explanation for quantitative experiments. As described in Sec. B.1.2, we do not rely on image captions but construct entirely unseen instructions during the evaluation stage. In contrast, other vision generalist models rely on instructions seen during training and incorporate image captions in their evaluation process. For clarification, we use Lumina-mGPT (Liu et al., 2024) as an example to illustrate this difference.

In the evaluation process for Lumina-mGPT, as shown in Table 1, all instructions follow this format: "Generate an image according to the provided image, and according to the following caption: {Image Caption},<|image|>". For the experiments in Appendix B.1, the instructions are "Depth estimation. <|image|>" for the *Depth Estimation* task, "Semantic segmentation. <|image|>" for the *Semantic Segmentation* task and "Surface normal estimation. <|image|>" for the *Surface Normal Estimation* task. Any alteration to the format of these instructions leads to model failure or significantly degraded its performance.

In the following, we directly illustrate the variety of unseen instructions we constructed for each evaluation sample through examples from quantitative experiments (cf. Fig. 33 ∼ 43).

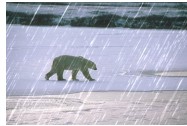

**Input Image**

**Unseen Explanatory Instruction**: "`The scene shifts from a wet and rainy environment to a calm and dry one by gradually reducing the intensity of the rain until it stops completely.`"

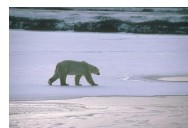

**Ground Truth**

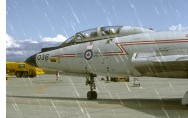

**Input Image**

**Unseen Explanatory Instruction**: "`Imagine the sky clearing up, the rain stopping entirely, and all moisture being removed from the scene, leaving the surface dry and the sky clear.`"

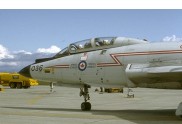

**Ground Truth**

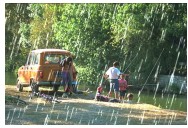

**Input Image**

**Unseen Explanatory Instruction**: "`To reach a scene where the weather turns fair, imagine the skies clearing and the rain ceasing, bringing out the sun. The ground dries up quickly as the rain evaporates, giving way to a brighter ambiance.`"

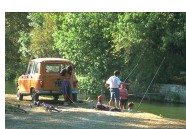

**Ground Truth**

*Figure 33.* Instruction-level zero-shot samples for quantitative experiments (*Deraining*).

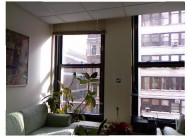

**Input Image**

**Unseen Explanatory Instruction**: "`The vibrancy intensifies, revealing a spectrum of colors that emphasize the angles and orientations of each surface in the scene.`"

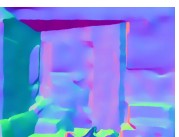

**Ground Truth**

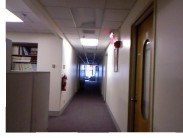

**Input Image**

**Unseen Explanatory Instruction**: "`Translate the visible structures into a range of bright colors reflecting orientation angles, enhancing variations across surfaces.`"

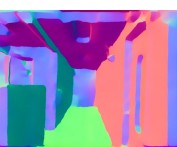

**Ground Truth**

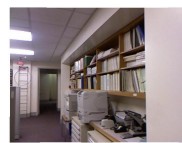

**Input Image**

**Unseen Explanatory Instruction**: "`Transform the visible elements into a vibrant array where distinct colors signify the orientation of surfaces.`"

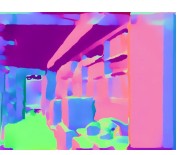

**Ground Truth**

*Figure 34.* Instruction-level zero-shot samples for quantitative experiments (*Surface Normal Estimation*).

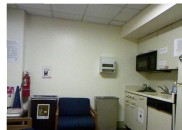

**Input Image**

**Unseen Explanatory Instruction**: "Reduce the color details and enhance the gradient of light to dark, emphasizing depth transitions. Apply a uniform grayscale filter across the entire scene, melding textures and patterns into smooth surfaces. Simplify distinct forms, softening and unifying borders to accentuate spatial differences."

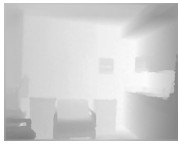

**Ground Truth**

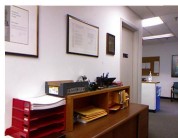

**Input Image**

**Unseen Explanatory Instruction**: "Introduce a gradient effect to the space, ensuring that the light source becomes the dominant feature. Simplify textures and colors to various shades of gray, emphasizing the depth and contours of objects. Focus on the relative positioning and shapes, creating a smooth transition from light to dark areas, highlighting the spatial arrangement."

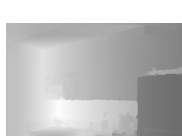

**Ground Truth**

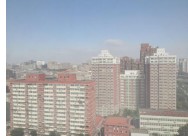

**Input Image**

**Unseen Explanatory Instruction**: "Convert all color data into grayscale, adjusting the brightness based on perceived depth, with closer objects appearing darker and farther ones lighter. Gradually diminish surface textures and fine details, focusing solely on the contours and relative positions of objects. Preserve the edges and outlines to enhance depth perception, emphasizing structural differences. Amplify the contrast between closer and more distant objects to create a clear sense of spatial arrangement."

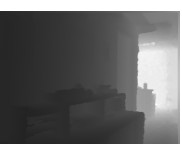

**Ground Truth**

*Figure 35.* Instruction-level zero-shot samples for quantitative experiments (*Depth Estimation*).

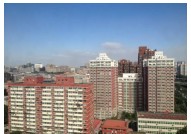

**Input Image**

**Unseen Explanatory Instruction**: "Enhance the clarity by reducing the haze, adjusting contrast and brightness levels for a sharper and more detailed view."

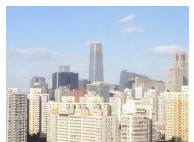

**Ground Truth**

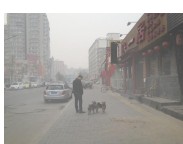

**Input Image**

**Unseen Explanatory Instruction**: "Increase the vibrancy and contrast while reducing atmospheric haze to reveal details, making sky and structures more defined."

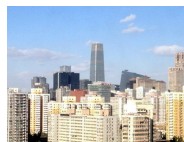

**Ground Truth**

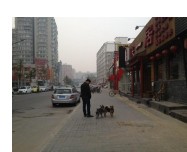

**Input Image**

**Unseen Explanatory Instruction**: "To enhance clarity and vibrancy, imagine each object gaining sharpness and colors becoming more pronounced. Gradually remove the grayish tone, revealing the street with increased contrast and brightness."

**Ground Truth**

*Figure 36.* Instruction-level zero-shot samples for quantitative experiments (*Dehazing*).

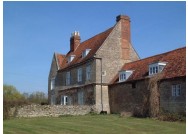

**Unseen Explanatory Instruction**: "Simplify the intricate details, focusing on large areas of color. Reduce the textures of the walls, roof, and garden to single, solid colors, ensuring a clear separation between different elements."

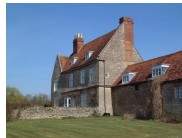

**Input Image**

**Ground Truth**

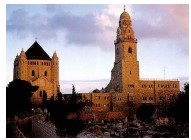

**Unseen Explanatory Instruction**: "Simplify the scene by identifying distinct regions such as the sky, structures, and foliage, assigning each a distinct color."

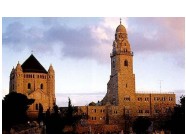

**Input Image**

**Ground Truth**

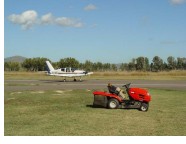

**Unseen Explanatory Instruction**: "Begin by abstracting detailed features into flat, vivid colors, transforming complex textures into simple hues. Convert the varied landscape elements into solid bands of color, maintaining only their fundamental positions and shapes. The sky turns a uniform light blue, and the plane becomes a defined mint green silhouette. The mower is simplified into a blue shape, with the seat becoming a highlight in red."

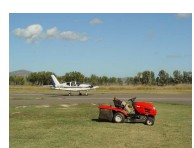

**Input Image**

**Ground Truth**

*Figure 37.* Instruction-level zero-shot samples for quantitative experiments (*Semantic Segmentation*).

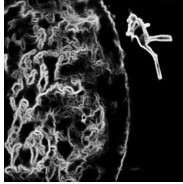

**Unseen Explanatory Instruction**: "Adding rich layers of color and texture to the outlines would allow for a return of the original setting's vibrancy. Explosive growth of texture from flat lines and edges would populate the space with marine life and details, enhancing the divers' surroundings and restoring their environment. Subtle gradients and vivid aquatic elements, like fish and the deep blue hues, re-emerge, filling the outlines with life."

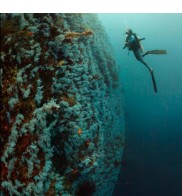

**Input Image**

**Ground Truth**

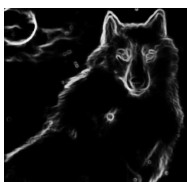

**Unseen Explanatory Instruction**: "Start by filling in the abstract lines with rich details, deepening the creases of the outline with fur textures and shadowed areas. Build up the dimensionality by adding layers of color, texture, and shading to reconstruct the depth in every feature. Soften the stark contrasts while adding fine details, such as the reflective light in the eyes and the nuanced light scattering of fur. Bring back the ambient background details, incorporating stars and subtle sky gradients to restore fullness to the scene."

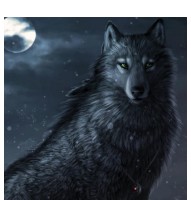

**Input Image**

**Ground Truth**

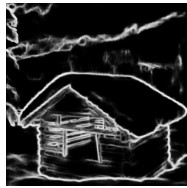

**Unseen Explanatory Instruction**: "Refill the outlines with authentic textures, colors, and shading. Introduce the interplay of natural lighting, allowing shadows and highlights to emerge once again over every visible surface. Rebuild the snowy forest backdrop, enhancing the dimensionality of the mountain scene with vibrant snow-covered elements under warm, diffuse sunlight. The cabin should regain the illusion of physicality, reflecting the interaction between the structure and the surrounding environment, including intricate snow details on the roof and ground."

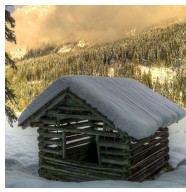

**Input Image**

**Ground Truth**

*Figure 38.* Instruction-level zero-shot samples for quantitative experiments (*HED-to-Image*).

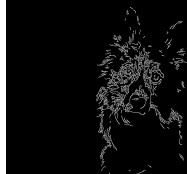

**Unseen Explanatory Instruction**: "Add colors, lighting, and fur texture gradually to the line-based outline, introducing gradients that replicate the realism of fur. Ensure soft lighting is integrated into the scene with attention to shading and depth on the dog's face and body."

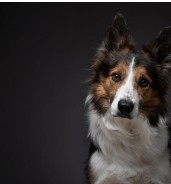

**Input Image**

**Ground Truth**

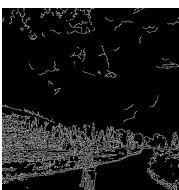

**Unseen Explanatory Instruction**: "Rebuild the scene by filling in all edges with rich color, shading, and texture. Reconstruct the orange clouds in the sky and provide depth to the environment by reintroducing lighting, perspective, and atmospheric effects, along with adding detail to individual objects like the child and bicycle."

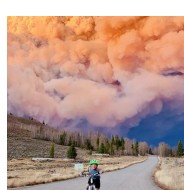

**Input Image**

**Ground Truth**

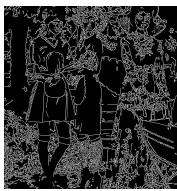

**Unseen Explanatory Instruction**: "Starting from abstract outlines, vibrant colors would need to emerge, filling in the spaces between the lines with rich textures, colors, and delicate lighting. Details like the lushness of surroundings and the intricate clothing details would have to grow back into the scene, building a sense of volume, dimensionality, and narrative where before there was only flat abstraction."

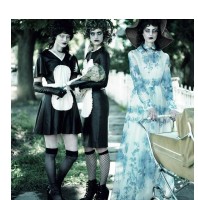

**Input Image**

**Ground Truth**

*Figure 39.* Task level & instruction-level zero-shot samples for quantitative experiments (*Canny-to-Image*).

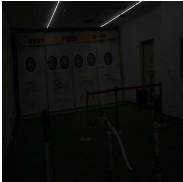

**Unseen Explanatory Instruction**: "To improve visibility and clarity, increase the brightness uniformly across the entire area. Next, adjust the contrast to differentiate objects and their surroundings, ensuring the whites and highlights stand out without losing shadows. Apply color correction to restore the vibrancy of dull tones, balancing any tints to make them appear natural. Enhance sharpness to recover fine details in textures, like the targets, arrows, and floor. Smooth out any noise that might emerge from the exposure increase, ensuring a clean and crisp finish while retaining natural lighting properties."

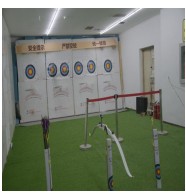

**Input Image**

**Ground Truth**

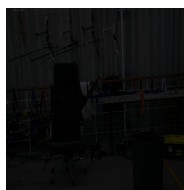

**Unseen Explanatory Instruction**: "Begin by enhancing the overall brightness of the scene to bring out hidden details. Gradually increase the contrast to differentiate the elements more clearly. Apply a color correction to amplify the muted tones, ensuring that the hues are vivid and lifelike. Finally, utilize a sharpening technique to accentuate the edges and textures, providing a more defined appearance."

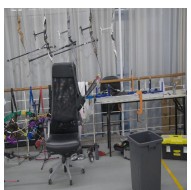

**Input Image**

**Ground Truth**

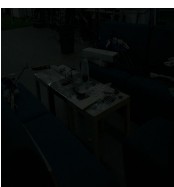

**Unseen Explanatory Instruction**: "Begin by adjusting the overall brightness of the scene to allow more light to penetrate the darker areas. Gradually increase the contrast to ensure that the highlights pop while maintaining some depth in the shadows. Employ techniques to amplify color saturation, bringing out the richness of hues that were previously subdued. Finally, apply sharpening filters to enhance the clarity of fine details that were not discernible in the original."

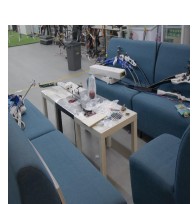

**Input Image**

**Ground Truth**

*Figure 40.* Task level & instruction-level zero-shot samples for quantitative experiments (*Low-light Enhancement*).

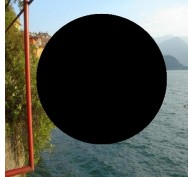

**Unseen Explanatory Instruction**: "`Remove the circular obstruction to reveal the previously hidden water and foliage, restoring the full view of the landscape.`"

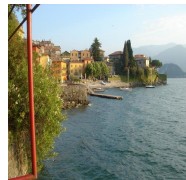

**Input Image**

**Ground Truth**

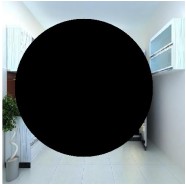

**Unseen Explanatory Instruction**: "`Reveal the hidden elements by removing the black circle, restoring the visibility of the entire kitchen setup.`"

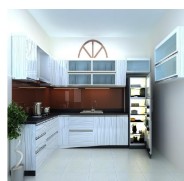

**Input Image**

**Ground Truth**

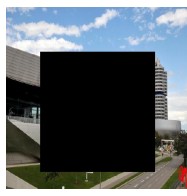

**Unseen Explanatory Instruction**: "`Remove the central black square to reveal the previously obscured parts of the architectural structure and landscape, restoring the full scene.`"

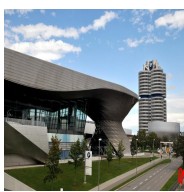

**Input Image**

**Ground Truth**

*Figure 41.* Task level & instruction-level zero-shot samples for quantitative experiments (*Inpainting*).

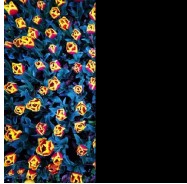

**Unseen Explanatory Instruction**: "`Expand the vertical floral display into a broader composition, ensuring the blooms are evenly spread across, with minimal visible soil.`"

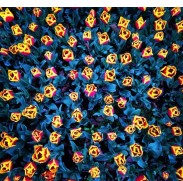

**Input Image**

**Ground Truth**

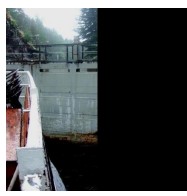

**Unseen Explanatory Instruction**: "`Expand the view horizontally to capture a wider perspective, incorporating additional elements from the periphery to create a broader context.`"

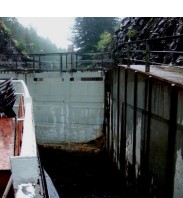

**Input Image**

**Ground Truth**

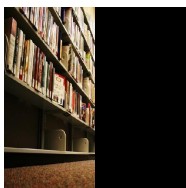

**Unseen Explanatory Instruction**: "`Expand the view by extending the horizontal dimensions on both sides to reveal more of the surrounding aisle. This creates a wider scene, replicating the perspective of a broad corridor lined with books extending into the distance.`"

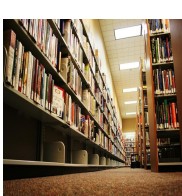

**Input Image**

**Ground Truth**

*Figure 42.* Task level & instruction-level zero-shot samples for quantitative experiments (*Outpainting*).

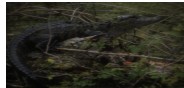 **Unseen Explanatory Instruction**: "Enhancing the details by reducing the blurriness, increasing sharpness, and adjusting the contrast to bring clarity to the scene." 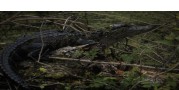

**Input Image**                                                     **Ground Truth**

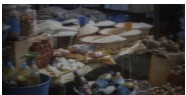 **Unseen Explanatory Instruction**: "Increase the sharpness and enhance the contrast to reveal clear details and make the colors more vibrant." 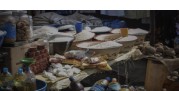

**Input Image**                                                     **Ground Truth**

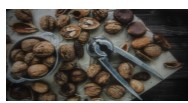 **Unseen Explanatory Instruction**: "Gradually refine the clarity by reducing any soft blur and enhancing details, while increasing contrast gently to make textures more pronounced." 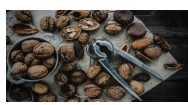

**Input Image**                                                     **Ground Truth**

*Figure 43.* Task level & instruction-level zero-shot samples for quantitative experiments (*Under-display Camera Image Restoration*).

## C. More Examples for Zero-shot Capabilities on Vision Tasks

In this section, we provide additional examples to demonstrate instruction-level and task-level zero-shot capabilities. The examples are generated using the model trained on the full DECVT. Note that all task-level zero-shot examples inherently fall under the category of instruction-level zero-shot examples as well. Additionally, we highlight the model's shortcomings based on these examples. For a more detailed exploration of limitations, please refer to Appendix D.

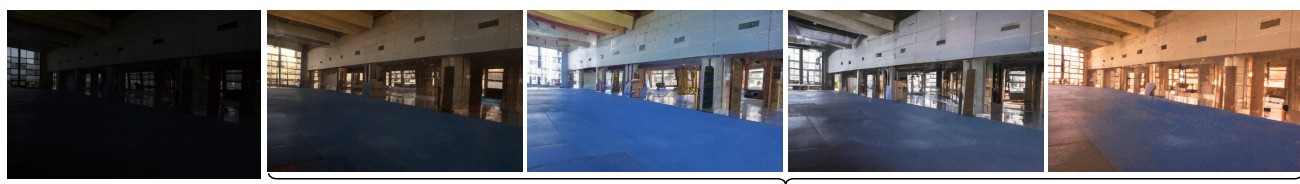

Input Image                                    Output Image

*Figure 44.* Examples of both task- and instruction-level zero-shot capabilities (*Low-light Enhancement*). Resolution: 448×448. Explanatory Instruction: "`Increase the overall brightness to reveal details in dark areas while preserving highlights. Adjust the contrast to enhance the brightness differences between regions, making the structures and textures more distinct. Optimize color saturation to make previously dull colors more vibrant, such as the blue on the floor becoming more prominent. Apply denoising to reduce noise commonly found in low-light images, improving the overall quality. Ensure the final image appears natural while retaining the authentic style of the scene.`" Limitations: Controlling the intensity of lighting enhancement through language instructions is challenging, often resulting in significant deviations in the output.

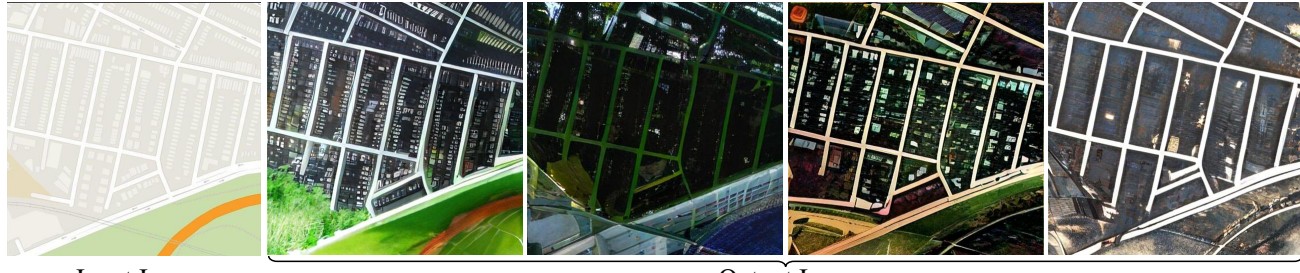

Input Image                                    Output Image

*Figure 45.* Examples of both task- and instruction-level zero-shot capabilities (*Map-to-Image*). Resolution: 448×448. Explanatory Instruction: "`Apply a process that introduces intricate details by simulating realistic textures, adding natural elements like trees and vegetation, and enhancing geometric shapes into more organic forms. Convert abstract layouts into visually accurate structures, enhancing color depth and including real-world spatial elements such as roof patterns and environmental surroundings.`" Limitations: The results still fall short when compared to real-world scenes.

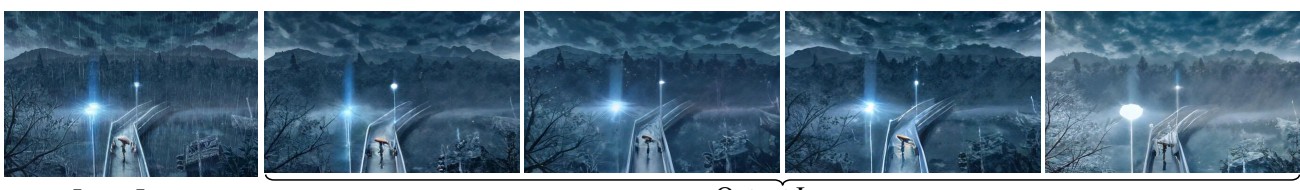

Input Image                                    Output Image

*Figure 46.* Examples of instruction-level zero-shot capabilities (*Deraining*). Resolution: 448×448. Explanatory Instruction: "`Slowly remove the rain falling from the sky in the image, still maintain the state of night, and the girl on the bridge is also still holding the umbrella, but readjust the light in the distance.`" Limitations: The model struggles to preserve smaller objects and environmental details.

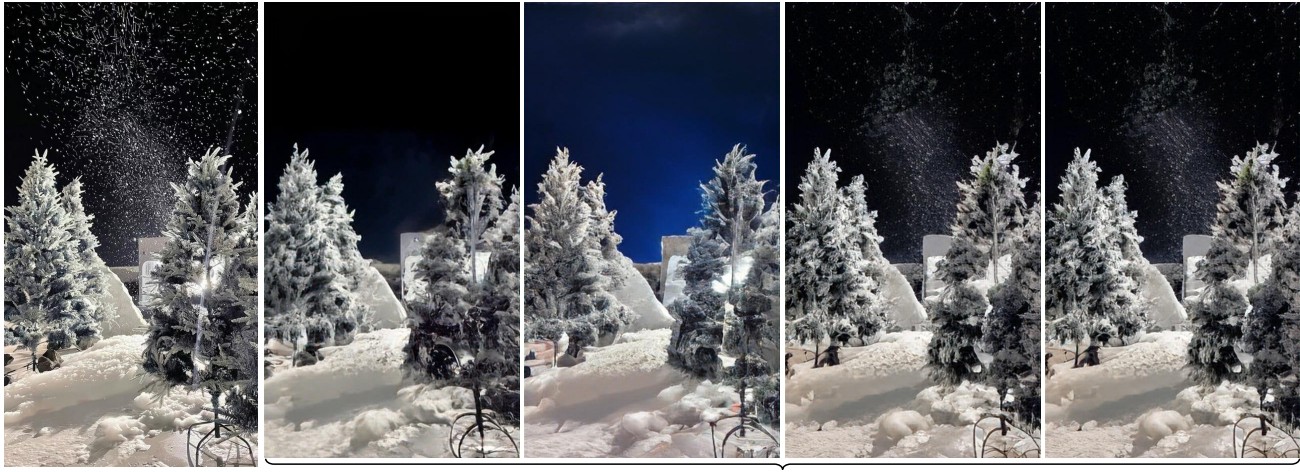

Input Image

Output Image

*Figure 47.* Examples of instruction-level zero-shot capabilities (*Desnowing*). Resolution: 448×448. Explanatory Instruction: "`Remove the falling snow from the sky in the image, keep the other objects and snow in the image, still keep it dark, but pay attention to the adjustment of light behind the tree.`" Limitations: The second generated image struggles to retain nighttime details, while the third and fourth images exhibit poor performance in removing snow from the sky. Additionally, attempting to remove snow from the ground simultaneously can result in significant distortions.

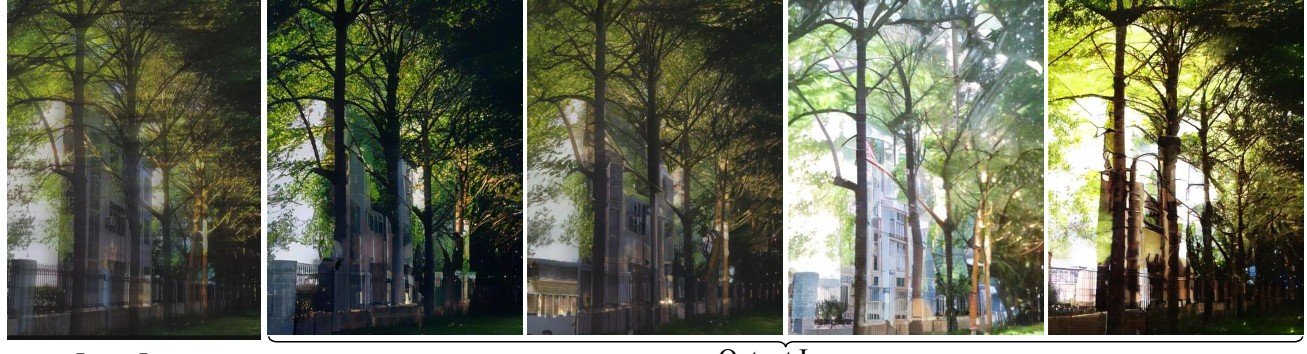

Input Image

Output Image

*Figure 48.* Examples of both task- and instruction-level zero-shot capabilities (*Deblurring*). Resolution: 448×448. Explanatory Instruction: "`The image shows noticeable multiple visual overlaps of trees and buildings. I would like to remove visual overlaps and restore a clear, sharp image without blurring. Do not alter the main content and pay attention to adjusting the light.`" Limitations: The success rate of guiding the model's task-level zero-shot capability through language instructions is relatively low.

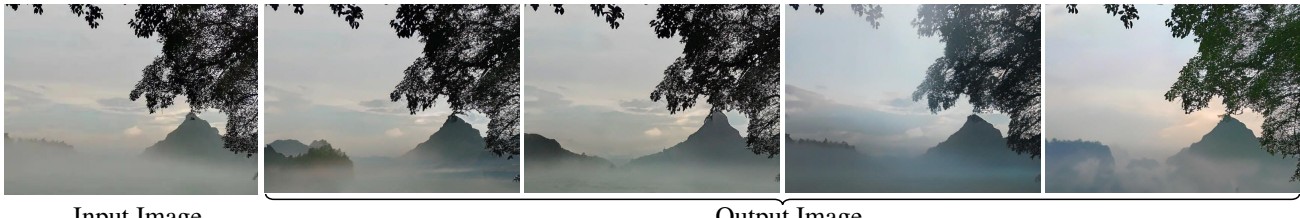

Input Image

Output Image

*Figure 49.* Examples of instruction-level zero-shot capabilities (*Dehazing*). Resolution: 448×448. Explanatory Instruction: "`Retain the distant clouds in the image while removing as much fog as possible. Attempt to restore the faintly visible sun in the distance, but ensure there is no strong sunlight. Focus on recovering the mountains and the nearby trees as much as possible.`" Limitations: It will cause distortions in certain objects.

In the following, we provide a few simple examples used for evaluation.

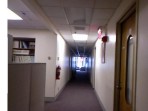

**Unseen Explanatory Instruction**: "Acknowledge the spatial structure and identify variations in light intensity, translating these into a gradient scale representing distances. Accentuate regions where light diminishes gradually, enhancing the perception of depth by dimming peripheral areas. Adjust the distribution of luminance to highlight the central vanishing point, converting detailed textures into smooth transitions of grayscale."

**Input Image**

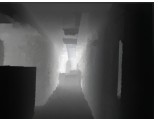 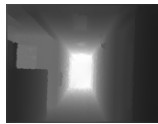

**Output Image**  **Ground Truth**

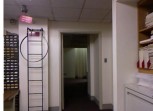

**Unseen Explanatory Instruction**: "Start by analyzing the spatial layout to identify key structural elements. Gradually obscure less relevant details in the periphery to focus primarily on central depth. Increase contrast between light and dark areas to enhance perception of distance. Transition the textures into smooth gradients to reflect variations in depth, with a focus on enhanced luminosity for regions that are further away."

**Input Image**

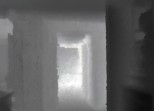 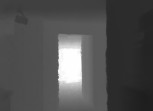

**Output Image**  **Ground Truth**

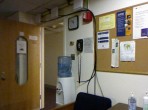

**Unseen Explanatory Instruction**: "Convert each region's color intensity to a grayscale value corresponding to its relative distance from the viewer, with nearer objects appearing lighter and those farther away darker. Gradually smooth transitions between these regions to reflect continuous depth variation. Remove textural details that do not affect perceived depth to create uniformity based on object proximity. Adjust overall brightness to highlight the spatial configuration without explicit texture representation."

**Input Image**

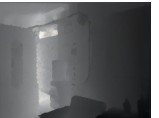 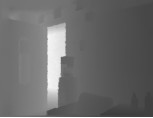

**Output Image**  **Ground Truth**

*Figure 50.* Examples of instruction-level zero-shot capabilities (*Depth Estimation*). Resolution: 448×448.

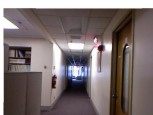

**Unseen Explanatory Instruction**: "Translate the visible structures into a range of bright colors reflecting orientation angles, enhancing variations across surfaces."

**Input Image**

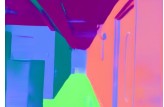 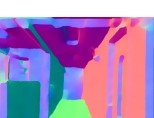

**Output Image**  **Ground Truth**

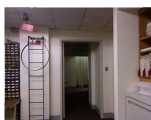

**Unseen Explanatory Instruction**: "Convert visual elements into a spectrum of colors that represent the directionality of surfaces, capturing the angles and orientations vividly."

**Input Image**

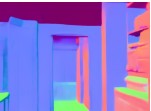 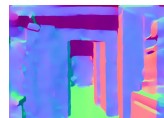

**Output Image**  **Ground Truth**

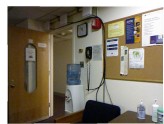

**Unseen Explanatory Instruction**: "Translate the scene into a colorful array to indicate surface orientations and angles."

**Input Image**

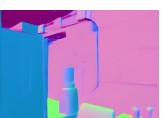 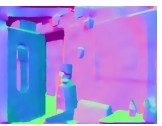

**Output Image**  **Ground Truth**

*Figure 51.* Examples of instruction-level zero-shot capabilities (*Surface Normal Estimation*). Resolution: 448×448.

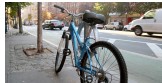 **Unseen Explanatory Instruction**: "Apply a pink color overlay to bicycles, completely matching their shapes." 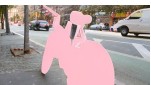 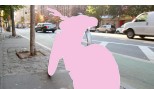

**Input Image**                                                                                        **Output Image**      **Ground Truth**

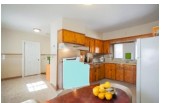 **Unseen Explanatory Instruction**: "Apply a solid grey color tint to fully cover one banana instance.Paint over each stove with a powderblue color." 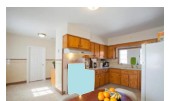

**Input Image**                                                                                        **Output Image**      **Ground Truth**

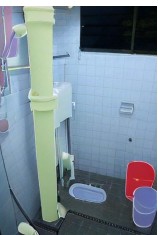 **Unseen Explanatory Instruction**: "Spectral_r is the reversed version of Spectral, transitioning through red, yellow, green, and blue. Based on the previously defined colors, help me complete the segmentation task below. Color all instances of bucket, toilet using Spectral_r colors, following their contours precisely." 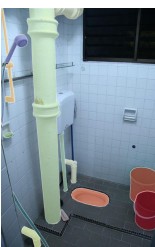

**Input Image**                                                                                        **Output Image**      **Ground Truth**

*Figure 52.* Examples of instruction-level zero-shot capabilities (*Semantic Segmentation*). Resolution: 448×448.

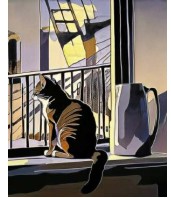 **Unseen Explanatory Instruction**: "Mark the cat-like creature in the picture with a yellow box." 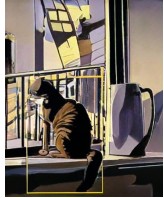 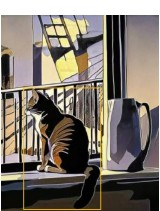

**Input Image**                                                                                        **Output Image**      **Ground Truth**

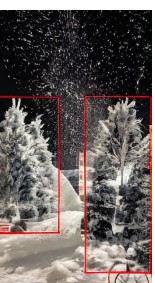 **Unseen Explanatory Instruction**: "Mark trees with snow in the picture with red boxes." 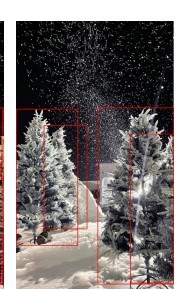

**Input Image**                                                                                        **Output Image**      **Ground Truth**

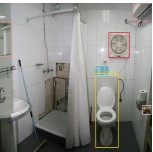 **Unseen Explanatory Instruction**: "For the following object categories, apply the corresponding bounding box colors: Encircle all toilet objects with a gold color border. Draw a thistle bounding box around all doorknob objects.Detect fan and apply a crimson bounding box around them. Detect air_conditioner, box color: palevioletred color. Detect all mirror and draw a darksalmon color bounding box.Detect toilet_tissue and apply a darkturquoise bounding box around them."

**Input Image**                                                                                        **Output Image**      **Ground Truth**

*Figure 53.* Examples of instruction-level zero-shot capabilities (*Object Detection*). Resolution: 448×448.

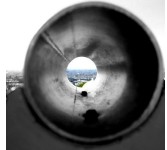

**Input Image**

**Unseen Explanatory Instruction**: "Capture the outline and prominent edges of the cylindrical object and its surroundings, simplify everything by removing textures and detailed surfaces, and emphasize only the contours and distinct features while rendering a higher contrast between light and dark regions with sharp shifts in tones. "

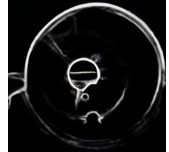 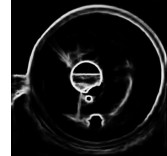

**Output Image**  **Ground Truth**

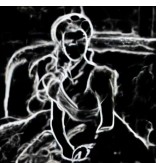 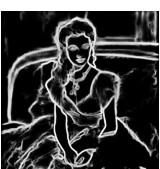

**Output Image**  **Ground Truth**

**Unseen Explanatory Instruction**: "The vibrant scene with multiple colors and details could be simplified into a monochrome representation. First, focus on defining the high-contrast areas between light and dark in a much starker, black-and-white way. Then, it's important to emphasize contours and significant edges, such as the lines around the face, the dress' folds, and the furniture's details, while downplaying softer gradients. Removing extraneous colors and textures leaves behind only the essential structural features that provide a more abstract, but recognizable silhouette and objects."

**Input Image**

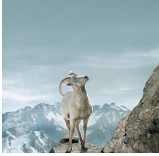

**Input Image**

**Unseen Explanatory Instruction**: "Begin by eliminating most of the intricate details and colors, transforming the vibrant elements into simplified outlines. Keep only the borders and defined structures, ensuring that the environment and figure take on an abstract form. Remove all texture, reducing the entire composition to minimal contrasting edges that define the shapes more than the details."

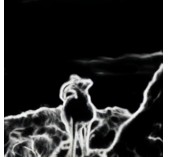 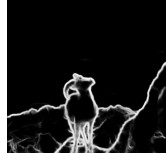

**Output Image**  **Ground Truth**

*Figure 54.* Examples of instruction-level zero-shot capabilities (*HED Boundary Detection*). Resolution: 448×448.

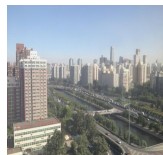

**Input Image**

**Unseen Explanatory Instruction**: "Gradually reduce atmospheric interference, allowing clearer visibility of buildings and sharpening the outlines. Enhance clarity and brightness to bring out the details within the cityscape, providing a crisper view."

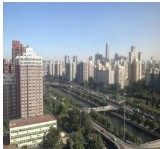 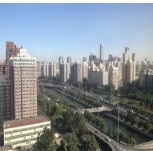

**Output Image**  **Ground Truth**

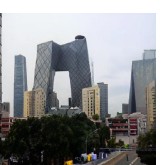 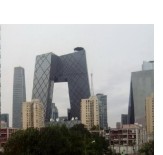

**Output Image**  **Ground Truth**

**Unseen Explanatory Instruction**: "Increasing the clarity by reducing haze, enhancing contrast, and deepening colors to give a sharper and more vibrant appearance to the scene."

**Input Image**

**Unseen Explanatory Instruction**: "To achieve clarity and vibrancy, adjust the brightness and reduce the foggy effect. Enhance the sharpness of the trees and structures, allowing their details to stand out against the clear blue sky."

**Input Image**

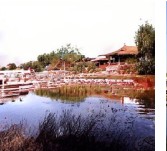 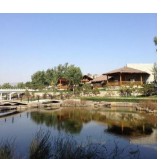

**Output Image**  **Ground Truth**

*Figure 55.* Examples of instruction-level zero-shot capabilities (*Dehazing*). Resolution: 448×448.

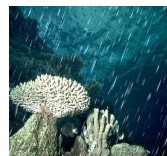

**Unseen Explanatory Instruction**: "Imagine a scenario where rainfall suddenly stops and the water settles, clearing up the scene to enhance visibility and eliminate rain streaks."

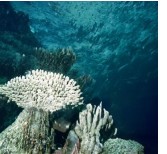 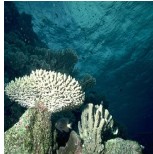

**Input Image**

**Output Image**    **Ground Truth**

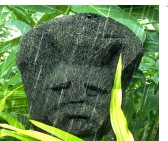

**Unseen Explanatory Instruction**: "Remove the raindrops and streaks, focusing on enhancing clarity and brightness to achieve a crisp and rain-free appearance in the environment."

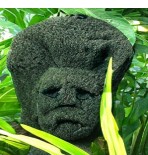 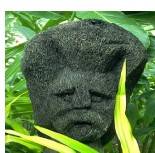

**Input Image**

**Output Image**    **Ground Truth**

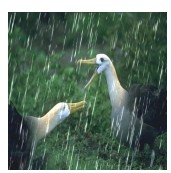

**Unseen Explanatory Instruction**: "Imagine the rainfall gradually lessening until the sky clears completely, leaving only the vibrant greenery and the birds in focus."

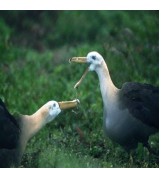 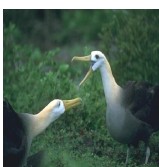

**Input Image**

**Output Image**    **Ground Truth**

*Figure 56.* Examples of instruction-level zero-shot capabilities (*Deraining*). Resolution: 448×448.

## D. Limitations and More Discussions

In this section, we provide more limitations and discussions of this work. Due to GPU resource constraints, we were unable to conduct additional validation experiments to address these issues. Therefore, we present only our reflections and insights based on the current results and our experiences.

1) As we have mentioned in Sec. 3 of the paper: "*While the top-k value is typically set to 5 for text generation in large language models (LLMs), we recommend setting the top-k to 2048 during the image generation stage*." This recommendation is also consistent with findings in Lumina-mGPT (Liu et al., 2024), where larger top-$k$ value lead to improved visual details. However, this phenomenon should not necessarily apply to all vision tasks, *i.e.*, in some vision tasks, larger top-$k$ value should not improve the generate results. For instance, in standard segmentation tasks controlled by colors and classes, the answer should be unique for most of the image, with only a few pixels potentially subject to ambiguity. While in such cases, lowering the top-k value still can not improve the model's ability to follow these instructions. Our intuition is that the current VQ-based training methods and the straightforward fine-tuning approach may have a significant impact on this issue.

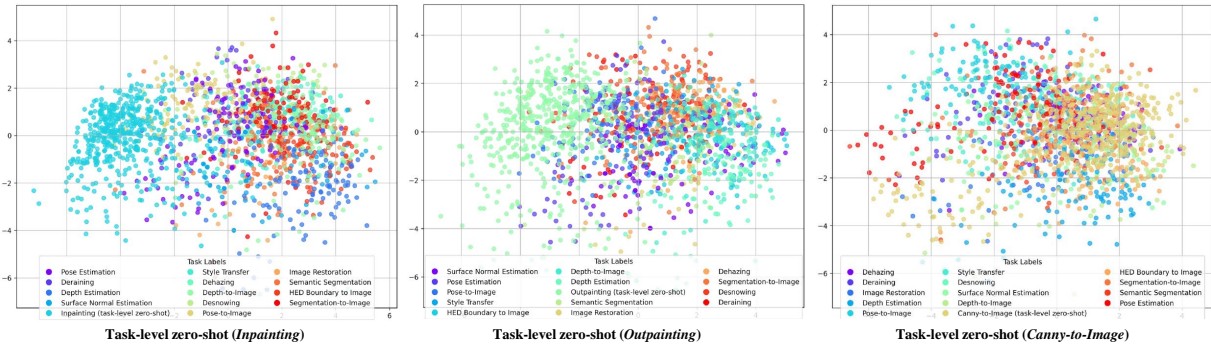

*Figure 57.* Latent space visualization of explanatory instructions.

2) When visualized in a reduced feature space, fixed task-specific instructions (*e.g.*, *Semantic Segmentation*) exhibit discrete clustering—a property that inherently ties task execution to rigid syntactic forms, potentially limiting generalization. In contrast, explanatory instructions span a continuous spectrum across tasks, which we identify as the primary driver of zero-shot generalization capability. To validate this, we extracted the text features of explanatory instructions for each task using the BERT (Devlin et al., 2019) model, and then applied PCA to reduce the high-dimensional features to two dimensions. For better visualization, we randomly sampled 100 features per task and plotted them in a scatter plot. As shown in Fig. 57, explanatory instructions avoid the tight clustering seen with task-specific instructions, instead forming overlapping distributions even across distinct tasks. This continuity—where task boundaries blur in the latent space—enables the model to generalize by exploiting semantic relationships between instruction formulations rather than relying on pre-defined task categories.

3) Experiments in this work are limited to image pairs, and we do not explore more complex data such as video or 3D data. Actually, in the early stage of our experiments, we attempted to construct explanatory instructions between different frames in videos. Unfortunately, for complex videos, even models like GPT-4o still produced significant errors within the descriptions (cf. Table 8). While for simpler videos (*e.g.*, those depicting weather changes), although the descriptions tend to be accurate, we feel that adding data with minimal changes between frames does not significantly contribute to testing the zero-shot capability of the model, which is the main goal of this work (we acknowledge that such data, with minimal changes, could potentially enhance the model's ability to control generation and follow instructions, but this is not the focus of this study). Nevertheless, we believe that collecting more complex video data could be beneficial for advancing vision task-level generalization. Such data would not only improve the model's scene control capabilities but also increase the diversity of task objectives that the model can understand. While these tasks could be treated as a form of editing, they surpass the capabilities of current editing models (*e.g.*, transformation from figure (a) $\sim$ (d) in Table 8).

4) In constructing the Dataset of Explanatory CV Tasks, we adhered as closely as possible to the principle that the provided instructions should avoid obvious inaccuracies. While GPT-4o exhibits one of the most advanced descriptive capabilities among existing models, it still faces challenges such as incomplete descriptions and occasional deviations. Although these data significantly enhance the model's ability to interpret a wide range of instructions and facilitate task-level generalization,

**Descriptions of image flow A:**

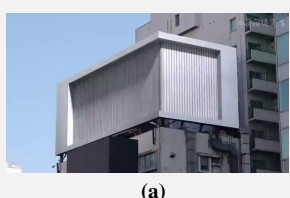 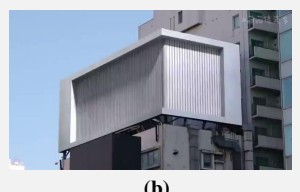 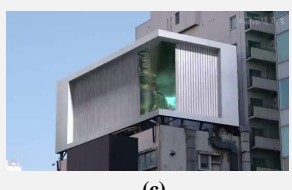

(a)                              (b)                              (c)

1. A closed structure with no visible contents.
2. The structure begins to reveal an internal object.
3. The internal object becomes partially visible.

**Descriptions of image flow B:**

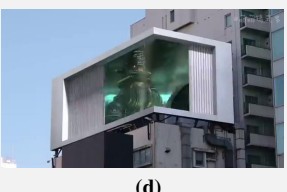 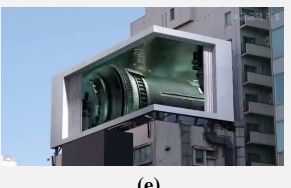 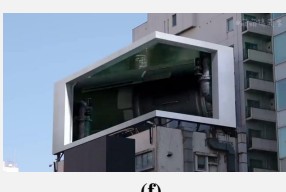

(d)                              (e)                              (f)

1. A partially revealed internal object within a structure.
2. The internal object becomes more visible.
3. The structure closes, hiding the internal object.

**Explanatory instruction from A to B:**
1. Start with a closed structure.
2. Gradually reveal an internal object.
3. Continue revealing more of the internal object until it is partially visible.
4. Progress to showing the internal object more clearly.
5. Finally, close the structure, hiding the internal object again.
**Explanatory instruction from B to A:**
1. Begin with a partially revealed internal object within a structure.
2. Gradually hide the internal object until it is no longer visible.
3. Ensure the structure is completely closed with no visible contents.
4. Open the structure slightly to start revealing an internal object.
5. Continue to reveal more of the internal object until it is partially visible.

*Table 8.* Video output sample from GPT-4o. Content that does not meet the description requirements is highlighted in red.

they also lead to certain trade-offs. Specifically, the inclusion of these data can reduce the stability of the model's outputs and compromise fidelity to the original image content. Furthermore, these data issues, including the presence of some low-quality datasets for the image editing tasks, may adversely affect the model's instruction-following capabilities.

5) Although the dataset construction and training approach described above have several limitations, we believe that the use of explanatory instructions can enhance the model's adaptability to complex instructions and objectives. We hypothesize that there exists an optimal level of instruction complexity for models. Up to this threshold, more detailed instructions may improve the model's understanding and performance. However, beyond this point, the model's ability to grasp the intended objective may begin to decline. Moreover, when instructions encompass overly broad objectives while the model lacks sufficient capacity to process and generalize this information, it may lead to performance degradation.

6) Due to budget constraints during the construction of the Dataset of Explanatory CV Tasks, a significant portion of the data was generated by directly instructing the model to output explanatory instructions, bypassing the generation of image captions. However, based on empirical observations, we recommend that the model generate image captions before producing explanatory instructions. As for GPT-4o, if the generated image captions exhibit substantial errors, the explanatory instructions are also likely to be inaccurate. Conversely, when the image captions are accurate, the explanatory instructions tend to be more precise. This approach helps to significantly reduce noise within the dataset.

7) Due to resource limitations, we only conducted our experiments on the vanilla token-based VLM with 7B parameter, and

no image-caption-based data were used for image generation during training. While this work demonstrates that explanatory instructions can enable zero-shot generalization at the vision task level, this ability remains unstable. Furthermore, on benchmark datasets, the performance of our model under zero-shot settings still lags behind task-specific models and some vision generalist models. This performance gap remains substantial compared to our expectations: as humans possess the ability to generalize across different tasks during the learning process and, based on these generalizations, can better accomplish existing tasks. We guess the primary reasons for the current limitations lie in the architecture of the model and data noise in the Dataset of Explanatory CV Tasks.

8) Just as with language understanding, visual cognition also vary from person to person. For various vision tasks, even when dealing with vision tasks related to a single image pair, different people may interpret and describe the images and tasks differently. Therefore, we think that the simple next-token prediction approach may be well-suited to those vision tasks that have already been generalized and defined by humans (*e.g.*, *Depth Estimation* and *Segmentation*). However, for more complex vision tasks, directly training through next-token prediction may not be the most appropriate approach, as task descriptions can vary widely and multiple valid answers may exist.

