# OpenReview forum: "Explanatory Instructions: Towards Unified Vision Tasks Understanding and Zero-shot Generalization"
_ICML.cc/2025/Conference — ICML 2025 poster_

### Official Review · Reviewer_7Dut · 2025-03-07

**Overall Recommendation:** 3

**Summary:**

This work proposed a large-scale explanatory instruction dataset to unify multiple CV tasks for AR VLM understanding and then generation. It uses VQ-VAE style tokens for vision and then an AR model to merge the 2 modality. It provides qualitative results on various tasks, showing the zero-shot capabilities.

**Claims And Evidence:**

Yes.

**Essential References Not Discussed:**

I think the VideoPoet work bears resemblance to what the authors want to achieve, using AR VLM to do some zero-shot tasks, but there was no mention of it.

**Experimental Designs Or Analyses:**

1. The model is trained for 2 epcohs. My impression is that most VLM or AR model either during pre-training or fine-tuning train for 1 epoch. Does the author justify the setup?

**Methods And Evaluation Criteria:**

Yes.

**Other Comments Or Suggestions:**

I think the captions of the images can be more explanatory for reading flow, but that is optional.

**Other Strengths And Weaknesses:**

Weakness:

1. It is common knowledge that an instruction method such as this paper proposed would be helpful to in the proposed task. So, I'm not sure if the novelty is not already understood or practiced.

2. There are not many quantitative results to objectively compare with others as a whole. (I know there are a few in the appendix.)


Strength:
1. The paper provided a large-scale synthetic dataset that integrates multiple CV tasks.

**Questions For Authors:**

So given the dataset is a bit like generated from GPT-4o, wouldn't it be like distilling ability from GPT? and resulting in a chicken-n-egg problem? What if there is no ChatGPT?

**Relation To Broader Scientific Literature:**

This work contributes to the areas of VLM understanding and generation, more specifically, showing a unified AR model can do multiple CV tasks and zero-shot generalization.

**Theoretical Claims:**

None in this work.

---

> ### Author Rebuttal · Authors · 2025-03-29
>
> Thank you for your feedback. We realize there may be some misunderstandings regarding both the dataset and idea presented in our paper. We hope the following responses will help clarify these points.
>
> **Experimental Designs**: *The model is trained for 2 epcohs.*
>
> **Response**: Thank you for your comments. Most large-scale VLM or AR models typically adopt only 1 epoch during pre-training, primarily due to the sheer volume of data available, which is usually sufficient for the model to generalize effectively. However, for SFT, it is reasonable to employ more than 1 epoch, since the dataset size is typically smaller and involves more complex tasks. Specifically, our dataset includes rich explanatory instructions covering diverse vision tasks, necessitating multiple exposures for the model to effectively learn and generalize. Empirically, we observed improved performance and better instruction comprehension when training for 2 epochs compared to just 1. We will clarify this reasoning to avoid confusion.
>
> **Weakness 1**: *It is common knowledge that an instruction method would be helpful to in the proposed task.*
>
> **Response**: We have to clarify that the vision task-level zero-shot generalization has not been clearly addressed by any prior work. Indeed, prior sota vision-language models do not demonstrate this capability to generalize across multiple distinct vision tasks. The idea is also considered highly innovative by Reviewers #v7hm, #Mwwt, and #yk8P. We notice that the reviewer might misunderstand the type of generalization we target. Specifically, we distinguish clearly between **task-level zero-shot generalization** and **data-level zero-shot generalization**, the latter of which has indeed been extensively studied in prior works.
>
> Data-level zero-shot generalization, exemplified by models like VideoPoet that the reviewer mentioned, refers to generating diverse outputs (e.g., images or videos) conditioned on various textual prompts within a single predefined vision task, such as text-to-image or text-to-video generation. Although VideoPoet achieves impressive zero-shot performance in video generation tasks, it still remains constrained to the single task of video generation and does not exhibit cross-task generalization capabilities. In contrast, our paper explicitly addresses the fundamentally different challenge of task-level zero-shot generalization, where the model is expected to understand and execute completely novel and diverse vision tasks solely based on explanatory instructions, without task-specific fine-tuning.
>
> Moreover, to facilitate and validate this task-level generalization, we have constructed a dataset consisting of approximately 12 million individual "input image → explanatory instruction → output" triplets. Unlike existing datasets, which typically contain predefined task-specific annotations, our dataset uniquely provides rich explanatory instructions that describe the objectives of diverse vision tasks explicitly. To the best of our knowledge, this dataset is the first and currently the only dataset specifically designed for studying and enabling unified task-level zero-shot generalization in vision tasks.
>
> **Weakness 2**: *There are not many quantitative results.*
>
> **Response**: We appreciate the reviewer raising this concern. To clarify, we have indeed provided extensive quantitative evaluations in the supplementary materials, covering 12 different tasks in total. For each task, representative visual examples are also included in Appendix Section C for further transparency.
>
> It is important to note that all baseline methods we compared against, with the exception of Lumina-mGPT, do not utilize instruction-level or task-level zero-shot settings. While Lumina-mGPT is closer to our method in principle, it still requires rigid and fixed-format prompts for each specific task, as explicitly discussed in lines 1546\~1551 of the paper, and it did not provide any quantitative results for both instruction-level or task-level zero-shot experiments. Thus, it does not truly operate in a fully zero-shot, instruction-driven manner as proposed by our method. Additionally, to the best of our knowledge, this paper is the first work to provide quantitative analyses under both instruction-level and task-level zero-shot settings.
>
> **Question**: *The dataset is like distilling ability from GPT?*
>
> **Response**: The reviewer seems misunderstood the construction of the dataset. Before the submission of this paper, the three GPT-4o models officially released by OpenAI, i.e., gpt-4o-2024-05-13, gpt-4o-2024-08-06, and chatgpt-4o-latest, did not possess image generation or image editing capabilities, let alone the vision task-level generalization capability to be addressed by our work. In the construction of this dataset, we only adopt gpt-4o to generate part of the explanatory instructions, while other explanatory instructions for terminological-based vision tasks are manually annotated.

---

### Official Review · Reviewer_yk8P · 2025-03-09

**Overall Recommendation:** 4

**Summary:**

This paper proposes a concept called “Explanatory Instructions” to move beyond the conventional limitations of computer vision (CV) tasks. The authors argue that the currently common terminological definitions (e.g., “semantic segmentation”) oversimplify the expression of CV objectives, limiting the model’s ability to achieve generalized zero-shot performance across tasks. To address this, they propose using detailed natural language instructions to describe transformations between input and output images in a large-scale dataset. The idea is that by exposing the model to these more expressive and fine-grained task descriptions, it learns to “understand” the underlying objectives and changes for each visual task. Built on an autoregressive vision-language model their experiments show that the method not only generalizes in zero-shot fashion to previously seen tasks but also demonstrates some ability to handle unseen tasks.

**Claims And Evidence:**

The primary claim of the paper is, by introducing Explanatory Instructions and constructing a corresponding large-scale dataset, the authors claim to achieve zero-shot generalization for both seen and unseen vision tasks.
The authors have presented numerous qualitative examples—such as generating reasonable outputs for previously unseen image-to-image transformation tasks—and some initial tests to illustrate zero-shot performance.
While the qualitative results are intuitive, more thorough quantitative evaluations or systematic comparisons with existing baselines would bolster the credibility of their claims about task-level zero-shot generalization.

**Essential References Not Discussed:**

While the paper covers recent VLM-related literature well, it would benefit from discussing more explicitly related previous work on task-level generalization in vision-language domains, such as [1] and [2].

[1] Bachmann R, Kar O F, Mizrahi D, et al. 4M-21: An any-to-any vision model for tens of tasks and modalities. NeurIPS, 2024: 61872-61911.
[2] Xiao B, Wu H, Xu W, et al. Florence-2: Advancing a unified representation for a variety of vision tasks. CVPR, 2024: 4818-4829.

**Experimental Designs Or Analyses:**

The key experimental setup is distinguishing between tasks included in the training set and tasks that remain unseen, testing for zero-shot performance on the latter. Most analyses focus on sample outputs and visual demonstrations of how the model handles various image editing and transformation tasks. While these examples highlight the model’s adaptability, a more fine-grained quantitative analysis, such as performance under different instruction styles or varied input modalities would provide additional insight into robustness and failure modes.

**Methods And Evaluation Criteria:**

I think the proposed methods and evaluation criteria in this paper are appropriate for addressing zero-shot generalization in vision tasks. The intuitive that define CV task objectives through detailed linguistic transformations from input images to outputs also sounds great.
-Methods: The authors adopt a standard autoregressive Transformer as the backbone for their vision-language generative model, converting both image and text inputs into discrete token sequences, and training via next-token prediction.
-Datasets and Evaluations: The authors constructed a large dataset, separating some tasks to remain unseen for zero-shot testing. This design helps assess whether the model can generalize to tasks not included in its training set.
From an application perspective, replacing fixed “task-name” categories with more flexible, instructive natural language is a promising approach. It also aligns with current instruction tuning trends in NLP. However, it seems that the evaluations make sense are primarily qualitative visualizations.

**Other Comments Or Suggestions:**

Authors can consider providing quantitative evaluation metrics (FID, CLIP scores, etc.) in the main paper.

**Other Strengths And Weaknesses:**

Other Strengths:
1.Substantial scale of dataset (12 million triplets) facilitating robust training.
2.Promising demonstration of task-level zero-shot generalization beyond conventional terminological boundaries.
3.Clearly articulated ideas and methodological innovations.

Other Weaknesses:
1.It seems that this paper could benefit from stronger baselines or comparative studies with existing methods.
2.This paper also lacks some in-depth analysis for the proposed Explanatory Instructions, details can refer to “Questions for authors” part.

**Questions For Authors:**

As mentioned in the weaknesses, I am more focused on some analytical issues regarding Explanatory Instructions.

1. Could Explanatory Instructions potentially exhibit clustering patterns? For instance, short editing instructions and fixed task-specific instructions (e.g., “Semantic Segmentation”) used in some VLMs might cluster around a few points when their features are extracted and visualized. I wonder whether different descriptions of the same task, when expressed via Explanatory Instructions, would demonstrate similar properties. If even for the same task, the representations via Explanatory Instructions in the evaluation dataset show significant bias. For example, wide dispersion in feature space. This could provide stronger evidence for the feasibility of task-level zero-shot generalization.

2. As real-world instructions can be vague or ambiguous. So if the instruction phrasing, length, style, or ordering changes, does this result in notable performance differences for image generation or understanding tasks?

If Explanatory Instructions show higher intra-task diversity and clear inter-task separation, it would empirically validate their ability to capture task objectives beyond rigid terminological definitions.

**Relation To Broader Scientific Literature:**

This work is building on vision-language models and closely tied to the current trend of creating generalist models for vision tasks. Recent methods, such as Lumina-mGPT and OmniGen, rely on the notion of “task tag + input-output” to unify tasks such as image generation, segmentation, depth estimation, etc. In contrast, this paper introduces more descriptive natural language explanations to capture the goal behind each task, potentially enlarging the task space and improving zero-shot feasibility.

**Theoretical Claims:**

The paper does not include formal theoretical claims or proofs, so there was nothing to verify in this regard.

---

> ### Author Rebuttal · Authors · 2025-03-30
>
> Thank you for your positive evaluation of our paper. Below are our responses to the concerns you raised.
>
> **Essential References Not Discussed**: *While the paper covers recent VLM-related literature well, it would benefit from discussing more explicitly related previous work on task-level generalization in vision-language domains.*
>
> **Response**: Thank you for your suggestions. We will discuss these works in our paper.
>
> \-------------------------------------------------------------------------------------------
>
> **Question 1**: *Could Explanatory Instructions potentially exhibit clustering patterns?*
>
> **Response**: Thank you for your insightful comment. Based on your suggestion, we analyzed 14 different **Terminological-based Vision Tasks** (including *Image Restoration*, *Deraining*, *Dehazing*, *Desnowing*, *Object Detection*, *Style Transfer*, *Depth Estimation*, *Surface Normal Estimation*, *Pose Estimation*, *Semantic Segmentation*, *HED Boundary to Image*, *Pose-to-Image*, *Depth-to-Image*, *Segmentation-to-Image*) to explore whether Explanatory Instructions exhibit clustering patterns.
>
> We extracted the text features of Explanatory Instructions for each task using the BERT model, and then applied PCA to reduce the high-dimensional features to two dimensions. For better visualization, we randomly sampled 100 features per task and plotted them in a scatter plot.
>
> Results can be found in the following anonymous link: https://anonymous.4open.science/r/ICML_Re-C752/instruction_features_pca_bert_all_tasks.png .Our findings show that **the Explanatory Instructions, when used to express vision tasks, do not exhibit the same clustering patterns as fixed task-specific instructions** (e.g., "Semantic Segmentation"). Fixed task-specific instructions tend to be highly discrete in nature, which may hinder the model’s generalization capability. On the other hand, Explanatory Instructions display a greater degree of continuity, which enhances the model’s ability to generalize across different vision tasks.
>
> Additionally, we further investigated the zero-shot performance of specific vision tasks during testing. Specifically, we plotted the Explanatory Instructions used during both the training and testing phases in the same manner. For instruction-level zero-shot, we focused on the Explanatory Instructions for the same task in both the training and testing phases. We provide 3 examples in the following anonymous link: https://anonymous.4open.science/r/ICML_Re-C752/Instruction_level_zero_shot_bert_pca_3_samples.pdf . For task-level zero-shot, we visualized all the Explanatory Instructions across the training and testing tasks. For the selected zero-shot vision task, we random sample 500 features. We also provide 3 examples in the following anonymous link: https://anonymous.4open.science/r/ICML_Re-C752/task_level_zero_shot_bert_pca_3_samples.pdf . The above results demonstrate that in both instruction-level zero-shot and task-level zero-shot scenarios, there is almost no overlap between the training set and the test set.
>
> We hope this analysis helps clarify the potential of Explanatory Instructions for enhancing task-level zero-shot generalization, and we appreciate your suggestion for further exploring clustering patterns.
>
> \-------------------------------------------------------------------------------------------
>
> **Question 2**: *As real-world instructions can be vague or ambiguous. So if the instruction phrasing, length, style, or ordering changes, does this result in notable performance differences for image generation or understanding tasks?*
>
> **Response**: We thank the reviewer for raising this important question. Indeed, real-world instructions can vary significantly in phrasing, length, style, and ordering, and such variations can impact model performance. To thoroughly address this concern, we have provided numerous examples of real-world instructions in Appendix C, explicitly illustrating a wide diversity in phrasing, length, style, and structure.
>
> In our validation experiments, the unseen explanatory instructions indeed include substantial variations in linguistic expression. We acknowledge that different descriptive language choices can lead to noticeable differences in the model's generation or interpretation outcomes. However, it is precisely this linguistic diversity that facilitates stronger generalization capabilities of our model. For instance, as discussed explicitly in Appendix B.1.4 (Fig. 32), when encountering categories not included in the training set (e.g., "broad-winged damselfly"), the model struggles to recognize the category name alone. Yet, it significantly benefits from alternative descriptive expressions (e.g., "the creature on the leaf"), highlighting the model’s improved interpretability under instruction diversity.
>
> To further illustrate this behavior, we provide additional examples at the following anonymous link: https://anonymous.4open.science/r/ICML_Re-C752/More_Examples.pdf .

---

> > ### Comment · Reviewer_yk8P · 2025-04-03
> >
> > After reading the review, I am satisfied with the author's response, therefore I maintain my decision of giving a score of four.

---

> > > ### Author Response · Authors · 2025-04-03
> > >
> > > Thank you for your feedback! We sincerely appreciate the time and expertise you've dedicated to evaluating our work.  We will implement all suggested improvements to enhance the academic rigor and presentation clarity in the final version.

---

### Official Review · Reviewer_Mwwt · 2025-03-10

**Overall Recommendation:** 4

**Summary:**

This paper proposes Explanatory Instructions to address the challenge of task-level zero-shot generalization in computer vision, inspired by the success of instruction-driven models in NLP. The authors hypothesize that conventional terminological task definitions (e.g., "semantic segmentation") limit models' ability to generalize, as they fail to convey task objectives intuitively. To overcome this, they introduce Explanatory Instructions—natural language descriptions of image-to-output transformations—and construct a Dataset of Explanatory CV Tasks (DECVT) with 12M "image-input → instruction → output" triplets. A token-based autoregressive vision-language model is fine-tuned on DECVT, demonstrating instruction-level (unseen instructions for seen tasks) and task-level (unseen tasks like Canny-to-Image) zero-shot capabilities. Experiments show improvements over baselines in image editing, generation, and low-level tasks, though performance gaps remain compared to task-specific models.

**Claims And Evidence:**

Most claims made in the submission are supported by clear and convincing evidence, e.g.,
1) Explanatory Instructions improve instruction-level and task-level zero-shot generalization.
2) DECVT enables models to handle diverse vision tasks via linguistic instructions.
3) The AR-VLM achieves competitive results on unseen tasks (e.g., Depth-to-Image) compared to vision generalist models.

**Essential References Not Discussed:**

Overall, the authors cite most of the major and emerging literature on multi-task vision models and multimodal pretraining.

**Experimental Designs Or Analyses:**

The authors make comprehensive experiments across 10+ tasks in Appendix B. During the evaluation for task-level zero-shot tasks, it seems that authors have excluded specific tasks during training, which make the zero-shot scores convincing. However, I think there need some discussions for failure cases.

**Methods And Evaluation Criteria:**

As the unified token-based AR-VLM architecture aligns with recent trends in multimodal modeling, this paper further constructed a systematic dataset of explanatory CV tasks, combining manual annotation and GPT-4o-generated instructions. Although the results in Table 5 ~ 7 still show significant gaps compared to state-of-the-art task-specific models, both the task-level and instruction-level zero-shot generalization for CV tasks shown in this paper are exciting.

**Other Comments Or Suggestions:**

None

**Other Strengths And Weaknesses:**

I just brief summarize some other strengths and weakness.

Strengths:
1)	Novel idea of Explanatory Instructions for both task-level and instruction-level generalization in computer vision.
2)	The proposed large-scale dataset is meaningful to the community, the dataset also encompasses various transformations and task types.
3)	Authors provide various discussions in the appendix.
4)	If validated further, this approach could be an important training paradigm for general-purpose multimodal models.

Weaknesses:
1)	Overreliance on GPT-4o for instruction generation introduces potential noise and biases.
2)	There lacks an explanation regarding the distribution of data across different tasks in the dataset.
3)	Zero-shot experiments can address a subset of unseen tasks, while the authors have also discussed in Section 5. More diverse or challenging tasks need exploration to confirm broader scalability.

**Questions For Authors:**

1)	The authors used GPT-4o for training data generation, especially the instructions. From my experience, although GPT-4o seems better than other VLMs as discussed in Appendix A, there still remains potential noise and biases in the generated instructions. How the authors deal with these noises, biases, and even inaccuracies?
2)	Section 2 provides a detailed account of the dataset composition, and the supplementary materials explain how the dataset was constructed. However, certain details remain unspecified. For instance, in the task-level zero-shot experiments, what proportion of the data is allocated to each task? And in the complete dataset, what is the ratio of each task pair?
3)	The examples in Figure 32 are quite helpful in illustrating the effectiveness of explanatory instructions. However, the authors only provided pair of examples, and it would be beneficial if they included more convincing samples.

**Relation To Broader Scientific Literature:**

The paper clearly situates its contributions within the broader scientific literature. It identifies a clear gap between NLP and CV regarding zero-shot generalization capabilities, acknowledging previous approaches such as Lumina-mGPT, OmniGen, and PixWizard. The paper builds convincingly upon prior NLP-inspired vision-language paradigms and emphasizes the uniqueness of explanatory-based instruction as a conceptual advance beyond existing terminological frameworks.

**Theoretical Claims:**

The paper does not introduce formal theorems or rigorous mathematical proofs, so there is no specific need for verification of theoretical correctness. Its principal contribution lies more in method design and dataset construction than in theoretical innovation.

---

> ### Author Rebuttal · Authors · 2025-03-30
>
> Thank you for your positive feedback on our paper. In the following responses, we have addressed the concerns you raised during the review process, and we hope these answers will resolve your questions.
>
> **Question 1 / Weakness 1**: *The authors used GPT-4o for training data generation, especially the instructions. From my experience, although GPT-4o seems better than other VLMs as discussed in Appendix A, there still remains potential noise and biases in the generated instructions. How the authors deal with these noises, biases, and even inaccuracies?*
>
> **Response**: We appreciate the reviewer highlighting this important issue. Although GPT-4o indeed provides superior capability compared to other VLMs (as detailed in Appendix A), the instructions it generates still contain noise, biases, or inaccuracies. To address these potential issues, we have implemented a data quality control procedure.
> To be specific, we noticed that GPT-4o can occasionally produce unstable or inaccurate explanatory instructions if it misunderstood the input images. To mitigate this, we simultaneously asked GPT-4o to generate captions corresponding to the images. We then employed CLIP to measure the semantic similarity between these captions and their associated images. Any generated data pair with a similarity score below 0.5 was automatically filtered out from the final dataset. Otherwise, we maintained the explanatory instructions. In addition, a portion of our dataset was manually annotated as described in Section A.1 (Ln, 718\~719), ensuring high-quality and bias-free explanatory instructions.
>
> \-------------------------------------------------------------------------------------------------------------------------
>
> **Question 2 / Weakness 2**: *Section 2 provides a detailed account of the dataset composition, and the supplementary materials explain how the dataset was constructed. However, certain details remain unspecified. For instance, in the task-level zero-shot experiments, what proportion of the data is allocated to each task? And in the complete dataset, what is the ratio of each task pair?*
>
> **Response**: Thank you for your suggestions. We clarify these details as follows:
>
> **Complete Dataset Composition**: Our full dataset consists of two components: Terminological-based Vision Tasks and Explanatory-based Vision Tasks. Specifically, the Terminological-based Vision Tasks component includes approximately 4M individual “input image → explanatory instruction → output” triplets for image editing tasks. For the other vision tasks within this component, each task has a varying number of triplets, ranging from a minimum of 0.05M to a maximum of 2M. In contrast, the Explanatory-based Vision Tasks component contains approximately 2M individual “input image → explanatory instruction → output” triplets.
>
> **Task-Level Zero-Shot Experiment Allocation**: As detailed in Section 4.2 (Ln. 320\~329), to **rigorously test task-level zero-shot generalization**, we deliberately removed data corresponding to the following tasks from the Terminological-based Vision Tasks component: *Image Restoration*, *Depth Estimation*, *Depth-to-Image*, *Surface Normal Estimation*, *Surface Normal-to-Image*, *HED Boundary Detection* and *HED-to-Image*. From the remaining tasks, we then constructed the training dataset for zero-shot experiments by randomly selecting 1M triplets from the Explanatory-based Vision Tasks component, 1M triplets specifically for image editing tasks, and another 1M triplets from other remaining vision tasks.
>
> \-------------------------------------------------------------------------------------------------------------------------
>
> **Question 3 / Weakness 3**: *The examples in Figure 32 are quite helpful in illustrating the effectiveness of explanatory instructions. However, the authors only provided pair of examples, and it would be beneficial if they included more convincing samples.*
>
> **Response**: Thank you for your suggestions. We have provided additional examples at the following anonymous link: https://anonymous.4open.science/r/ICML_Re-C752/More_Examples.pdf .

---

### Official Review · Reviewer_v7hm · 2025-03-15

**Overall Recommendation:** 3

**Summary:**

This work proposes a new method for fine-tuning visual tasks, explanatory Instructions. Inspired by the work on text instruction fine-tuning, the authors aim to explore whether there is a generalization phenomenon in instruction fine-tuning for visual tasks. Therefore, they construct pure-text instructions for different visual tasks and use a large vision model, specifically Lumina-mGPT-7B-768-Omni, to verify the effectiveness of the constructed instruction dataset. The authors observe both intra-task generalization ability and cross-task generalization ability. Meanwhile, this article proposes an Explanatory Instructions dataset.

**Claims And Evidence:**

NA

**Essential References Not Discussed:**

NA

**Experimental Designs Or Analyses:**

NA

**Methods And Evaluation Criteria:**

NA

**Other Comments Or Suggestions:**

NA

**Other Strengths And Weaknesses:**

I really really like the idea proposed in this paper. However, the discussion on zero-shot task generalization in this paper is clearly insufficient. If my concerns can be addressed and the paper can be further improved, I will raise my score. Otherwise, I will lower my score after the rebuttal stage.

Strengths:
The idea put forward in this paper, namely that zero-shot generalization can be observed across different vision tasks, is an important finding. The publicly available dataset Explanatory Instructions will also make a significant contribution to the research community and accelerate the process of achieving true unification in visual tasks.


Weaknesses:
1. There are only relatively subjective case analyses as experimental results. It is unclear whether they have been cherry-picked, and there is no quantitative measurement of the true capability of solving tasks generalized in the zero-shot setting.
2. I am not sure if a well-trained stable diffusion model for image generation can also solve these problems.
3. I am curious about the cross-task generalization ability of the initial VLM model, Lumina-mGPT-7B-768-Omni, without any training. Does task generalization come from the base model or from instruction fine-tuning?

**Questions For Authors:**

1. I am curious about the cross-task generalization ability of the initial VLM model, Lumina-mGPT-7B-768-Omni, without any training. Does task generalization come from the base model or from instruction fine-tuning?

2. What if the base model is a text + image-to-image model, such as Stable Diffusion (SD)?

3. Are there any quantitative results? Case-level arguments are hardly convincing enough for me to determine whether the theory you proposed is truly reliable.

4. It is necessary to further demonstrate whether this cross-task generalization ability is inherent in the base model itself or is stimulated by the task fine-tuning method. It is recommended to use multiple base models for proof.

**Relation To Broader Scientific Literature:**

NA

**Theoretical Claims:**

NA

---

> ### Author Rebuttal · Authors · 2025-03-29
>
> We greatly appreciate your recognition of the idea presented in our paper. Below are our responses to the concerns you raised.
>
> **Question 1 & 4 / Weakness 3**: *I am curious about the cross-task generalization ability of the initial VLM model, Lumina-mGPT-7B-768-Omni, without any training. Does task generalization come from the base model or from instruction fine-tuning? It is necessary to further demonstrate whether this cross-task generalization ability is inherent in the base model itself or is stimulated by the task fine-tuning method. It is recommended to use multiple base models for proof.*
>
> **Response**: Thank you for your valuable suggestions. We would like to clarify that, in our experiments on analysing the **zero-shot capabilities on unseen vision tasks**, the base model used for initialization was **Lumina-mGPT-7B-768** (cf. 300~301 of the paper, it is a text-to-image generator). This base model **has no multi-task capabilities beyond text to image generation**, so it cannot perform other vision tasks on its own.
>
> Thus, any cross-task generalization observed comes from our instruction fine-tuning, not from an inherent ability of the pre-trained model. We explicitly discuss this in Appendix B.2 (lines 1546–1551), confirming that Lumina-mGPT-7B-768-Omni by itself does not generalize to unseen tasks – the generalization emerges only after training with our explanatory instructions.
>
> \-------------------------------------------------------------------------------------------------------------------------
>
> **Question 2 / Weakness 2**: *What if the base model is a text + image-to-image model, such as Stable Diffusion (SD)?*
>
> **Response**: Thank you for your suggestions. Using a stronger or more flexible foundation model (e.g. more advanced DiT model) could indeed further improve the overall performance. Our primary goal, however, was to validate the methodology of unified vision task-level generalization rather than to chase state-of-the-art results on any single base model. Furthermore, traditional stable diffusion model can only generate images but can not generate text. Although directly using stable diffusion (text + image-to-image) can also adopt such vision task-level zero-shot capability (following settings in Section 4.2 of the paper), to enhance the scalability of the model, i.e., enabling the model to simultaneously generate text, images, and other multimodal data, we therefore adopted the vanilla AR-based VLM introduced in Chameleon and Lumina-mGPT.
>
> \-------------------------------------------------------------------------------------------------------------------------
>
> **Question 3 / Weakness 1**: *Are there any quantitative results? Case-level arguments are hardly convincing enough for me to determine whether the theory you proposed is truly reliable.*
>
> **Response**: Thank you for your suggestions. Actually, we have provided **quantitative results in Appendix B (Ln. 1375\~1510)** of the paper. In particular, Table 6 and Table 7 in Appendix B reports results on 5 instruction-level zero-shot tasks (the instructions are not seen during training) and 5 task-level zero-shot tasks (both the task and instructions are not seen during training). **Notably, all methods used for comparison do not use instruction-level or task-level zero-shot settings except Lumina-mGPT.** However, Lumina-mGPT requires rigid and fixed format prompts as input, as we had discussed in Ln. 1546\~1551 of the paper:
> In the evaluation process for Lumina-mGPT in Table 6, all instructions follow this format: "``Generate an image according to the provided image, and according to the following caption: {Image Caption},<|image|>``". For the experiments in Appendix B.1, the instructions are "``Depth estimation. <|image|>``'' for the *Depth Estimation* task, "``Semantic segmentation. <|image|>``'' for the *Semantic Segmentation* task and "``Surface normal estimation. <|image|>``'' for the *Surface Normal Estimation* task. Any alteration to the format of these instructions leads to model failure or significantly degraded its performance.
>
> In addition, following the suggestions from Reviewer #yk8P (Question 1), we further demonstrate that under both the instruction-level and task-level zero-shot settings, there is a significant divergence between the training and test samples (visualizations for instruction-level zero-shot can be found in the following anonymous link: https://anonymous.4open.science/r/ICML_Re-C752/Instruction_level_zero_shot_bert_pca_3_samples.pdf , visualizations for task-level zero-shot can be found in the following anonymous link: https://anonymous.4open.science/r/ICML_Re-C752/task_level_zero_shot_bert_pca_3_samples.pdf). This provides additional evidence for the model’s generalization capability.

---

> > ### Comment · Reviewer_v7hm · 2025-04-04
> >
> > Thanks for the author's response. Some of my concerns have been addressed and clarified, yet others persist. The matter that concerns me is whether the method presented in the paper possesses sufficient generalization ability to be applied across other base models. For example, when considering the Supervised Fine-Tuning (SFT) for large language models, it is applicable to all such LLMs. Making a claim of zero-shot generalization is a significant assertion. The authors need to conduct more in-depth investigations to determine whether the instruct data is the crucial factor or if a powerful base model is the key determinant. Otherwise, misleading conclusions may ensue. Consequently, I will maintain my score unchanged.

---

> > > ### Author Response · Authors · 2025-04-05
> > >
> > > Thank you for your additional response. We are concerned that some details may not have been clearly expressed, so please allow us to provide a brief supplementary explanation:
> > >
> > > **1. Baseline Model Setup:**
> > > In our paper, we employ a fundamental AR-based vision-language model for experiments, which is built upon a basic large language model (LLM) augmented with a VQ-GAN for image encoding and decoding.
> > >
> > > **2. Lumina-mGPT-7B-768:**
> > > This model is trained under the aforementioned architecture and exhibits text-to-image generation capabilities. However, it does not inherently possess the ability to perform complex visual tasks. We use this model to directly demonstrate that Explanatory Instructions can enable vision task-level zero-shot generalization (as shown in Section 4.2).
> > >
> > > **3. Lumina-mGPT-7B-768-Omni:**
> > > This model, based on the same foundational architecture, can handle some visual tasks but remains limited (as discussed in Ln. 1546\~1551). Specifically, it only performs fixed visual tasks when provided with rigid, fixed-format language prompts. We conducted Supervised Fine-Tuning (SFT) on this model and showed that even a simple SFT can endow the model with both instruction-level and task-level zero-shot capabilities. Quantitative results demonstrating these abilities are provided in Appendix B.1, while Appendix C contains qualitative examples based on this SFT model.
> > >
> > > **4. Controlled Experiments and Contributions:**
> > > Through these controlled experiments, we have demonstrated that Explanatory Instructions are instrumental in eliciting both instruction-level and task-level zero-shot generalization. We also acknowledge that these zero-shot capabilities are partly influenced by the pre-trained visual encoder/decoder, an aspect we discuss further in Section 5.
> > >
> > > We understand your concerns regarding the claim of zero-shot generalization. Precisely because we were concerned that a powerful pre-trained model might obscure our assessment of zero-shot capability, we deliberately chose the aforementioned basic model for our validation experiments. We hope these clarifications can address your concerns. Once again, thank you for your valuable feedback and your positive comments on our idea.

---

### Decision · Program_Chairs · 2025-05-01

**Decision:**

Accept (poster)

**Comment:**

This paper proposes Explanatory Instructions to address the challenge of task-level zero-shot generalization in computer vision. All reviewers unanimously acknowledged that the methodology proposed by the authors exhibits a high degree of novelty, supported by comprehensive experimentation and substantiated content. Following extensive deliberation, the concerns raised by the reviewers have been adequately addressed by the authors.  Therefore, I recommend accepting this submission.